# MIXTURE OF COGNITIVE REASONERS: MODULAR REASONING WITH BRAIN-LIKE SPECIALIZATION

**Badr AlKhamissi**[1]   **C. Nicolò De Sabbata**[1]   **Greta Tuckute**[2,3]   **Zeming Chen**[1]

**Martin Schrimpf**[*,1]   **Antoine Bosselut**[*,1]

[1]EPFL   [2]Brain and Cognitive Sciences at MIT   [3]Kempner Institute at Harvard University

## ABSTRACT

Human cognitive behavior arises from the interaction of specialized brain networks dedicated to distinct functions, such as language, logic, and social reasoning. Inspired by this organization, we propose Mixture of Cognitive Reasoners (MICRO): a modular, transformer-based architecture post-trained with a curriculum that induces functional specialization across experts. Concretely, we partition the layers of a pretrained language model into four expert modules aligned with well-studied cognitive networks in the human brain. MICRO offers three key advantages over standard language models. (1) The specialized experts are interpretable and causally meaningful—ablating a module causes substantial drops on benchmarks requiring its specialized domain. (2) MICRO's behavior can be dynamically steered at inference time by routing tokens to particular experts (e.g., favoring social over logical reasoning), enabling fine-grained control over outputs. (3) MICRO outperforms or matches comparable baselines on both machine-learning reasoning benchmarks (e.g., GSM8K, BBH) and alignment to human behavior (CogBench), while maintaining interpretability. Taken together, cognitively grounded functional specialization yields models that are both more human-like and more human-interpretable.[1] [2]

## 1 INTRODUCTION

Neuroscience research suggests that distinct brain regions support language, reasoning, social cognition, and other cognitive functions (Saxe & Kanwisher, 2003; Kanwisher, 2010; Fedorenko et al., 2024). In contrast, the internal organization of Large Language Models (LLMs) is largely unstructured. While certain units or subnetworks show selective activation (Zhang et al., 2022; 2023; Bayazit et al., 2023; AlKhamissi et al., 2025a; Wang et al., 2025), such specialization is implicit and difficult to interpret or control. Motivated by this discrepancy, we propose a model architecture that explicitly incorporates specialization. On the machine learning (ML) side, such designs hold great potential for improving interpretability and controllability; on the cognitive science side, they provide a framework toward formulating testable computational hypotheses about how the relative contributions of different brain networks support complex behavior. To this end, we propose the Mixture of Cognitive Reasoners (MICRO), a class of modular language models that partition computation across brain-inspired expert modules.

The MICRO architecture partitions each layer of a pretrained language model into four experts, each designed to mirror a major cognitive network in the human brain: language (Fedorenko et al., 2011), logic (multiple demand; Duncan, 2010), social reasoning (theory of mind; Saxe & Kanwisher, 2003), and world knowledge (default mode; Gusnard et al., 2001). To provide the model with the inductive bias needed to learn this partitioning and cohesively integrate these experts, we design a three-stage

---

[*]Equal Supervision
[1]Code, data and models available at cognitive-reasoners.epfl.ch
[2]Demo available at huggingface.co/spaces/cognitive-reasoners

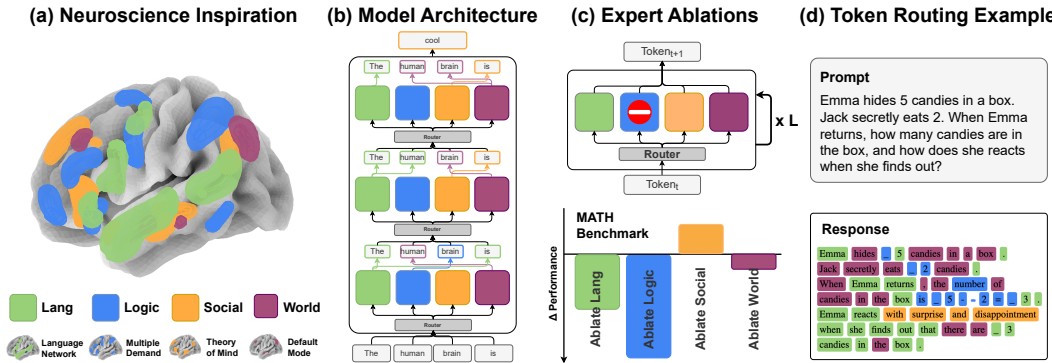

Figure 1: **Brain-Inspired Modular Language Model. (a)** Illustration of major cognitive networks in the human brain. **(b)** Our proposed Mixture of Cognitive Reasoners (MICRO) architecture. The MICRO architecture partitions each transformer block into four expert modules corresponding to analogous brain networks; a router assigns each token to an expert at every layer (i.e., assignments can vary across layers and tokens). **(c)** Illustration for causal steering via mechanistic ablations: removing a module shifts behavior and degrades domain-relevant performance. **(d)** Token-level routing on a sample prompt shows semantically coherent expert usage.

curriculum that uses lightweight training in the first two stages to sequentially (1) specialize the experts to mirror cognitive networks, and (2) bias a router to use certain experts for particular types of inputs (e.g., the logic expert for mathematics problems). The final training stage of this curriculum uses this now inductively-biased architecture to perform large-scale supervised finetuning.

Our results demonstrate that MICRO's architecture and training procedure induce interpretable specialization across these experts. This is evidenced by routing patterns and their correlations with human judgments (§5.1) and by causal ablations, which show dramatic drops in performance on reasoning categories when their corresponding experts are removed (§5.2). Moreover, the semantic behavior of these experts parallels the specialization of brain networks: (1) functional localizers used to recover brain-like mechanisms in LLMs (AlKhamissi et al., 2025a) identify the relevant experts in MICRO (§5.3), and (2) large MICRO models achieve high behavioral alignment scores on COGBENCH (Coda-Forno et al., 2024), a human behavioral benchmark, relative to two critical controls trained on the same data: (i) a mixture-of-experts model without induced brain-like specialization (MOB) and (ii) a non-modular dense transformer (DENSE) (§5.4). Finally, we find that MICRO's performance matches or exceeds these baselines (§5.5), indicating that interpretable and controllable specialization can be achieved without sacrificing overall performance.

## 2 BACKGROUND & RELATED WORK

### 2.1 NEUROSCIENCE MOTIVATION

Our design follows evidence that human cognition emerges from interacting, specialized brain networks. Cognitive neuroscience has mapped this modular organization by measuring how strongly different regions engage when people perform specific cognitive tasks (Kanwisher, 2010). We align MICRO's architecture with four core cognitive networks as shown in Figure 1(a). We summarize the functions of these networks below and their relevance to our modeling approach.

**The Language Network.** The *Language* expert mirrors the human language network, which comprises a set of left-lateralized frontal and temporal regions that selectively respond to linguistic input over perceptually matched non-linguistic stimuli (e.g., lists of nonwords; Fedorenko et al., 2010). These regions are highly specific to language, showing minimal activation during tasks such as arithmetic or music perception (Fedorenko et al., 2012; 2011), and their disruption can lead to selective language deficits (aphasia) without impairing general reasoning capabilities (Varley et al., 2005).

**The Multiple Demand Network.**   The *Logic* expert mirrors the Multiple Demand (MD) network, which spans bilateral regions and is activated across diverse cognitively demanding tasks such as difficult math problems, with stronger responses for higher difficulty levels (Duncan & Owen, 2000; Fedorenko et al., 2013). It correlates with fluid intelligence (Woolgar et al., 2010).

**The Theory of Mind Network.**   The *Social* expert mirrors the Theory of Mind (ToM) network, which is centered in the bilateral temporo-parietal junction and medial prefrontal cortex. This network supports reasoning about beliefs, intentions, and mental states (Gallagher et al., 2000; Saxe & Kanwisher, 2003; Saxe & Powell, 2006). It is robustly recruited across both verbal and non-verbal tasks involving perspective-taking and indirect communication (Koster-Hale & Saxe, 2013).

**The Default Mode Network.**   The *World* expert mirrors the Default Mode Network (DMN), which is active during rest and internally directed thought such as self-reflection, memory recall, and mental simulation (Gusnard et al., 2001; Buckner et al., 2008; Buckner & DiNicola, 2019). Centered in medial prefrontal and parietal regions, the DMN integrates information over long timescales, supporting discourse- and event-level processing across sentences or episodes (Ferstl & von Cramon, 2002; Jacoby & Fedorenko, 2020), in contrast to the shorter temporal window of the language network (Blank & Fedorenko, 2020).

## 2.2   Modular Language Models

In parallel with advances in cognitive neuroscience, recent years have seen growing interest in modular language models as a way to promote specialization, mitigate interference, and improve out-of-distribution generalization (Pfeiffer et al., 2023; Zhang et al., 2025). One major line of work centers on Sparse Mixture-of-Experts (MoE) architectures (Shazeer et al., 2017), with approaches ranging from curating domain-labeled datasets to train (Gururangan et al., 2022) or prompt (Si et al., 2023) domain-specific experts, to frameworks such as ModuleFormer (Shen et al., 2023), which introduce load-balancing and concentration losses to encourage modular specialization without explicit domain labels. Other modular approaches extend to multimodal integration (Liu et al., 2023; Swamy et al., 2023; Ye et al., 2023) or to disentangling representations by domain or language for multilingual and domain-specific applications (Pfeiffer et al., 2020; 2022; Zhong et al., 2022; Al-Maamari et al., 2024). In contrast, MICRO is, to our knowledge, the first modular language model explicitly designed to induce brain-like specialization, aligning experts with well-studied cognitive networks.

## 2.3   Brain-Inspired Models

Recent studies have shown that some models achieve strong alignment with activity in the human language network (Schrimpf et al., 2021; Toneva & Wehbe, 2019; Caucheteux & King, 2022; Aw et al., 2023; Tuckute et al., 2024a; AlKhamissi et al., 2025b). To further improve brain alignment, researchers have begun to integrate biologically inspired principles into model design—drawing from structures like the visual cortex hierarchy (Kubilius et al., 2019; Dapello et al., 2020; Spoerer et al., 2020), and the spatio-functional organization of the brain (Margalit et al., 2024; Binhuraib et al., 2025; Rathi et al., 2025).

# 3   The Mixture of Cognitive Reasoners Framework

## 3.1   Model Architecture

To build MICRO, we begin with a pretrained transformer-based backbone. For each layer, we clone the entire transformer block $N$ times, where $N$ corresponds to the number of experts intended for specialization, in a similar spirit to parameter upcycling (Komatsuzaki et al., 2023; Zhang et al., 2024). Then, we initialize a MLP-based router that assigns each token to a single expert. To maintain computational efficiency and a comparable number of active parameters to the original model, we use top-1 routing akin to Fedus et al. (2022). We refer to this architecture as *mixture-of-blocks* (MOB), distinguishing it from the more common mixture-of-experts (MOE), which restricts experts to FFN layers with shared attention. Importantly, we focus on MOB in the main paper because it induces clear functional specialization in all models, as reflected by lower router entropy and

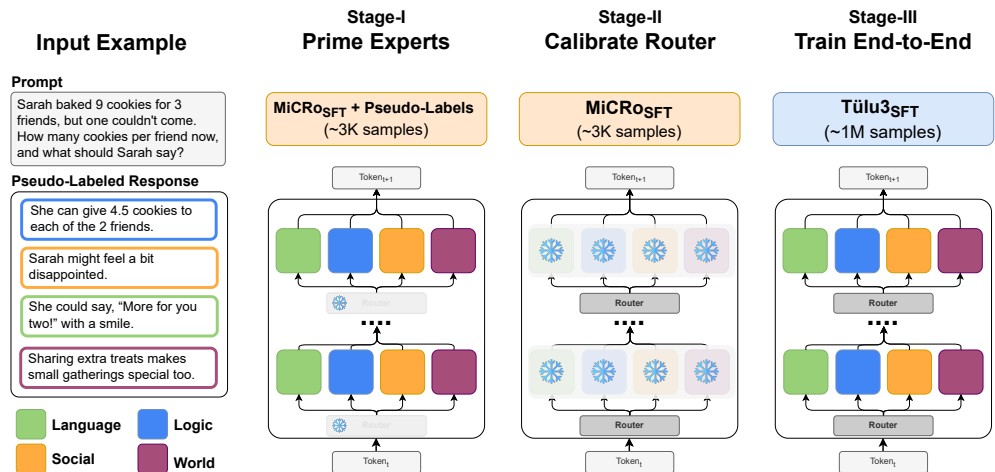

Figure 2: **Training Curriculum for Inducing Specialization.** Our brain-inspired Mixture of Cognitive Reasoners (MICRO) model contains four experts per layer, each aligned with a distinct cognitive network in the brain. In Stage-I, we train only the experts using a small, curated dataset MICRO$_{\text{SFT}}$ (see example on the left), providing each expert with an initial inductive bias. In Stage-II, we freeze the whole model and train the router on the same dataset to learn expert selection. In Stage-III, we finetune the entire model end-to-end on a large-scale instruction tuning dataset.

domain-consistent routing patterns, whereas MOE does not exhibit the same effect at specific scales (see Appendix H.1). Results for MICRO-MOE variants on reasoning benchmarks in Appendix H.2.

## 3.2 TRAINING CURRICULUM FOR INDUCING SPECIALIZATION

We induce functional specialization in MICRO experts using a three-stage training curriculum (see Figure 2). The first two stages use a small, curated dataset (MICRO$_{\text{SFT}}$) to provide targeted inductive biases, allowing specialization to emerge and solidify during the final full-scale training stage.

**Stage 1: Inducing Specialization.** In the first stage, we train the experts on a small dataset of $M = 3055$ examples (described below), each crafted to reflect the functional domain of a specific expert (Section 2.1). We denote this dataset as MICRO$_{\text{SFT}} = \left\{ (x_{i,1:T_i}, r_{i,1:T_i}) \right\}_{i=1}^{M}$, where each input sequence $x_i$ contains $T_i$ tokens, and $r_i$ provides token-level routing labels. Each label $r_{i,t} \in \{1, \ldots, N\}$ assigns the $t$-th token to one of the $N$ experts. This stage focuses solely on training the expert parameters using a next-token prediction loss. Tokens attend to all preceding tokens in the sequence regardless of which expert processed them using the key and value representations produced by the same expert. However, only tokens that are assigned to the expert in question continue to be processed through the feed-forward network. The same setup is applied in the next training stages, with the only difference that the router assigns the tokens to the experts.

**Stage 2: Calibrating Router.** Next, we freeze the whole model and train only the routers on the same dataset MICRO$_{\text{SFT}}$. The objective remains next-token prediction. Given the initial expert specialization from Stage 1, the router now learns to assign tokens to the most suitable expert. To encourage smoother transitions and more robust routing decisions, we use a soft mixture of the top-2 experts per token, which we found to be more effective than top-1 routing during this phase.

**Stage 3: End-to-End Supervised Finetuning.** Finally, we finetune the entire model end-to-end on a full instruction-tuning dataset, TÜLU-3 (Lambert et al., 2024), which consists of 939k examples. Even though this phase constitutes the majority of the training budget, we observe that the functional specialization seeded by the small MICRO$_{\text{SFT}}$ dataset is largely preserved (see Appendix J). Moreover, the experts continue to improve on tasks aligned with their initial domains, demonstrating that early inductive biases can lead to meaningful and lasting functional decomposition.

**Constructing the MICRO$_{\text{SFT}}$ Dataset.** To build MICRO$_{\text{SFT}}$ for inducing expert and router specialization, we first selected 19 existing reasoning datasets corresponding to the cognitive domains

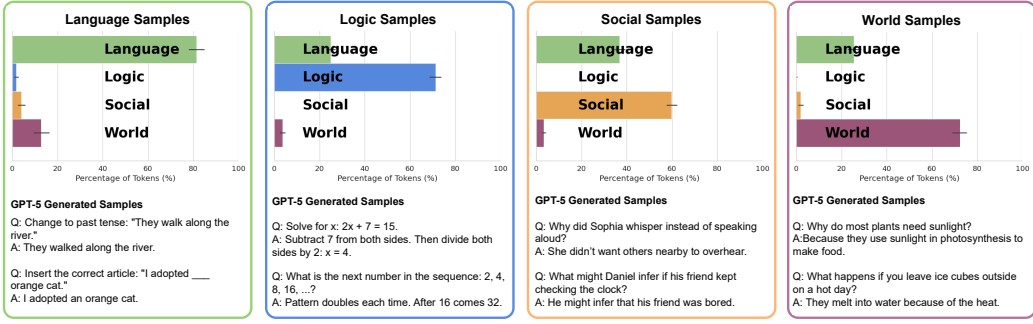

Figure 3: **Semantically Meaningful Routing Across Experts.** Token routing patterns in MICRO-LLAMA-1B. Each bar indicates the proportion of tokens routed to a given expert across layers, with variance shown across sentences (n=50). The model exhibits clear domain-specific specialization consistent with the intended brain-inspired organization. For example, social cognition samples are routed to the social expert, while arithmetic tasks are routed to the logic expert. We find that the language expert is consistently activated in the early layers (see Appendix C for layer-wise routing plots and results from additional models). Two random samples are shown below each subplot.

of our non-language experts, ensuring that each group of datasets spanned a diverse range of functions known to engage the corresponding brain networks. From each of the three sets, we randomly sampled 1,000 examples and used OpenAI's O1 model (Jaech et al., 2024) to generate detailed, step-by-step responses for each input. We then pseudo-labeled each sentence in the generated reasoning chains by prompting GPT-4O (Hurst et al., 2024) to assign it to one of the four experts. The tokens within each sentence inherit the corresponding expert label, which is used for deterministic routing in Stage 1. Details of the datasets are provided in Appendix A.

## 4 EXPERIMENTAL SETUP

We post-train five models of varying scales from three different families under our MICRO framework, in order to assess the generalizability of our method and identify the conditions under which it fails. Specifically, we use LLAMA-3.2-{1B, 3B} (Dubey et al., 2024), SMOLLM2-{135M, 360M} (Allal et al., 2025), and OLMO-2-1B (OLMo et al., 2024). Due to space constraints, we present the results of LLAMA-3.2-{1B, 3B} in the main paper while providing the full results for the remaining models in Appendix F. Each model is first finetuned for two epochs on the curated MICRO$_{SFT}$ dataset (Stages 1 and 2), followed by one epoch of end-to-end training using the TÜLU-3 dataset (Lambert et al., 2024), as described in Section 3.2. We use next token prediction as the only learning objective in all training stages, with the loss masked on the input tokens. We use an effective batch size of 32 and the AdamW optimizer across all stages. The learning rate follows a linear schedule, warming up over the first 3% of training to a peak of $2 \times 10^{-5}$, then decays linearly for the remainder of training. This schedule is applied for each stage separately.

**Reasoning Benchmarks.** We evaluate on four widely used reasoning benchmarks: GSM8K (Cobbe et al., 2021), MATH (Hendrycks et al., 2021b), BBH (Suzgun et al., 2022), and MMLU (Hendrycks et al., 2021a). Evaluation follows zero- or fewshot settings as detailed in Appendix F.

**Behavioral Benchmarks.** We evaluate alignment to human behavior using COGBENCH benchmark (Coda-Forno et al., 2024), which provides 10 metrics from 7 cognitive psychology experiments. These metrics capture how participants (or models) complete tasks that are designed to disentangle different behavioral strategies. Examples include *Directed Exploration*, *Meta-Cognition*, and *Risk Taking*. We refer readers to Coda-Forno et al. (2024) for a detailed description of the tasks.

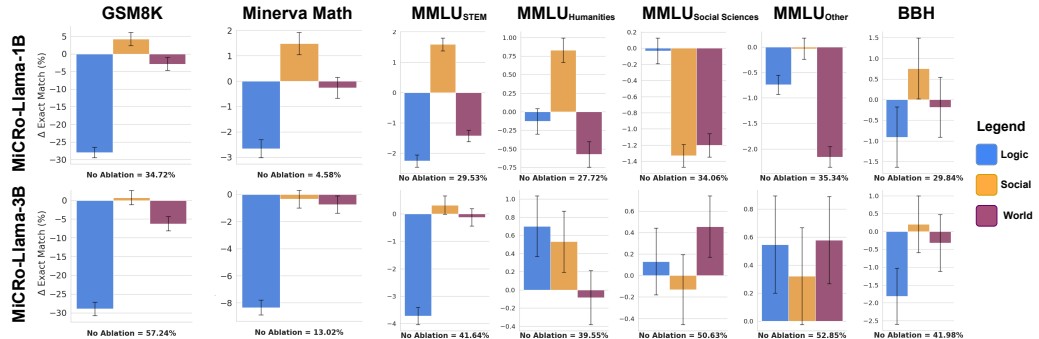

Figure 4: **Expert Ablations Reveal the Causal Contributions of Specialized modules.** Top and bottom panels show results for MICRO-LLAMA-1B and MICRO-LLAMA-3B. Removing the Logic expert causes large drops on MATH and GSM8K, while removing the Social expert yields slight gains. For MMLU and BBH, results indicate that some group of subtasks rely on distinct experts, whereas others draw on overlapping contributions. Additional models in Appendix E.

## 5 RESULTS & ANALYSIS

Our results unfold in two parts. First, we ask whether brain-like specialization emerges under our training curriculum, analyzing routing behavior, correlations with human judgments, causal ablations to test the functional contributions of those experts, and whether neuroscience experiments used to identify brain networks also identify the corresponding experts in our models. Second, we ask how this specialization influences alignment with human behavior and reasoning performance.

### 5.1 ROUTER PATTERNS ARE INTERPRETABLE AND CONSISTENT WITH HUMAN JUDGMENTS

**Token Routing Per Expert.** We first verify that our model routes tokens to the most relevant expert module, analogous to how specialized brain networks are selectively engaged by specific stimuli. Figure 3 shows the routing behavior of MICRO-LLAMA-1B, revealing clear domain-specific specialization. To generate test inputs, we sampled 50 question–answer pairs using GPT-5 prompted with descriptions of the four brain networks (prompt provided in Appendix C). Results for the other MICRO variants are consistent and reported in Appendix C. There, we also show routing patterns on reasoning benchmarks, where tokens are directed to experts consistent with the benchmark's domain. Finally, layer-wise analyses in Figures 13 & 14 reveal a hierarchical organization: earlier layers focus on linguistic grounding, while deeper layers increasingly delegate to domain-specific experts—an organization that emerged without being enforced by the training procedure and that parallels evidence from cognitive neuroscience (Fedorenko et al., 2024).

**Correlation with Human Judgments.** We evaluate model–human correspondence using 1,000 six-word sentences from Tuckute et al. (2024b), each annotated with human ratings across several behavioral dimensions (e.g., mental state content, grammaticality). These annotations were collected independently of our routing framework. We find that router probabilities correlate with the corresponding human judgments: for example, the social expert's selectivity aligns with ratings of mental state content ($r = 0.7$). Full results are provided in Appendix D.

### 5.2 EXPERTS ARE CAUSALLY MEANINGFUL

**Validation of Functional Experts via Ablations.** Figure 4 illustrates how expert ablations reveal the causal contributions of specialized modules to task performance. By selectively removing individual experts, we can directly test whether their specialization is functionally necessary for different domains. For example, on MATH and GSM8K, ablating the *Logic* expert causes a substantial drop in accuracy, confirming its central role in numerical reasoning. In contrast, removing the *Social* expert slightly improves performance, suggesting it plays a detrimental role in these tasks. For broader benchmarks such as MMLU, which span multiple subdomains, we report results for each

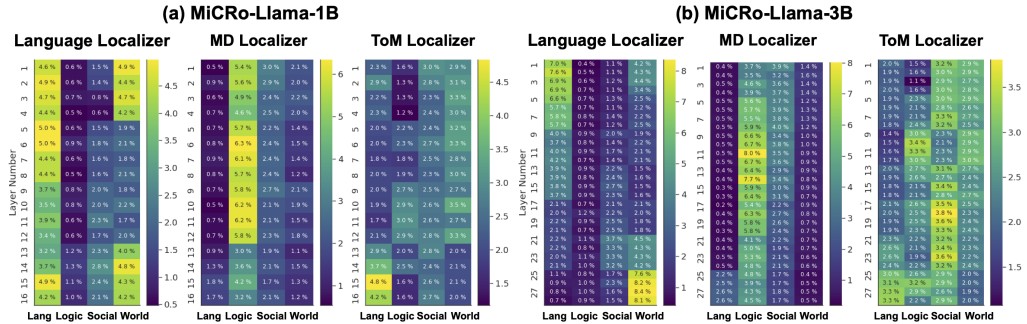

Figure 5: **Neuroscience Localizers Recover Functionally Specialized Experts.** (a) MICRO-LLAMA-1B and (b) MICRO-LLAMA-3B. For each model, we apply three neuroscience-inspired localizers—Language, Multiple Demand (MD) and Theory of Mind (ToM)—to examine the selectivity of localized units across experts and layers. Each plot shows the percentage of units in each expert of each layer that belongs to the top-10% selective units in the whole model.

subcategory separately. Performance drops after ablating the corresponding experts indicate that these clusters depend on distinct functional modules. Still, not all subtasks within a category align neatly with a single cognitive domain, and some require overlapping contributions, such as BBH. We show in Appendix E the effect of removing the language expert, which causes a significant drop on all benchmarks, along with additional ablation results on other models.

**Steering Model Behavior at Test-Time.** Our results demonstrate that test-time ablations can steer expert behavior, with social responses emerging when only the social expert is active and logical reasoning dominating when only the logic expert is retained. Qualitative examples in Appendix K.

### 5.3 NEUROSCIENCE LOCALIZERS REVEAL FUNCTIONAL EXPERT SPECIALIZATION

Neuroscientists rely on localizer experiments to identify the brain regions associated with specific functional networks, as their precise locations can vary across individuals. This raises a natural question: can we apply these established neuroscience localizers to identify the corresponding expert modules in our model? If so, this would provide further support for the hypothesis that our experts are functionally analogous to their associated brain networks.

To investigate this, we adopt the methodology of AlKhamissi et al. (2025a), which has been used to localize the language network, the multiple demand network, and the theory of mind network in LLMs. We apply these localizers to our MICRO models to test whether they can recover the corresponding expert modules. Figure 5 shows the percentage of units in each expert of each layer that belongs to the top 10% of selective units across the whole model, similar to what is done in the brain (Lipkin et al., 2022). The results show that language selectivity, as defined by the language localizer, favors the language expert at early layers while favoring the world expert at later layers for both models. The multiple demand localizer successfully favors the logic expert in both models. In contrast, ToM localization is less effective at isolating units within the social experts, but improves with scale, suggesting that ToM ability must emerge before it can be localized. One other possible reason for this is the limited size of the ToM stimulus set, which includes only 10 contrastive pairs, in contrast to 240 for language and 100 for multiple demand. This small sample may lack the robustness needed to reliably localize ToM-selective units (Jamaa et al., 2025).

### 5.4 STRONG ALIGNMENT TO HUMAN BEHAVIOR

Having established that our MICRO models exhibit human-like specialization (§5.1) that is causally linked to task performance (§5.2), we next examine whether they better align to human behavior compared to two baselines: one without brain-like specialization (MOB) and one without any modularization (DENSE). Both models are post-trained on a mixture of $2\times$ MICRO$_{SFT}$ and $1\times$ TÜLU-3 matching the total amount of data used in the MICRO training curriculum.

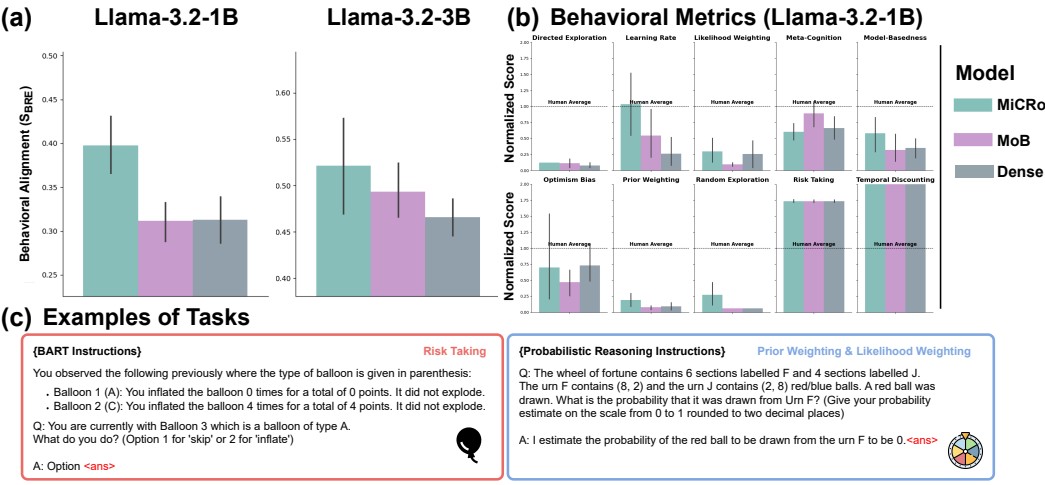

Figure 6: **Alignment with Human Behavior on COGBENCH. (a)** Average similarity score ($S_{\text{BRE}}$) across 10 behavioral metrics, showing that MICRO-LLAMA models achieves superior alignment compared to their MOB and Dense baselines. **(b)** Human-normalized scores for each metric separately across the three models. **(c)** Example inputs from two of the seven classical psychological experiments verbalized for LLM evaluation following COGBENCH.

Figure 6 presents the results on COGBENCH, evaluating alignment with human behavior. Unlike the original paper, which predicts answers via autoregressive generation, we pick the option with the highest log-probability for multiple-choice tasks to avoid invalid generations. Each experiment is run with five random seeds. Metrics are normalized such that random = 0 and human = 1. To quantify overall alignment, we introduce the bounded relative error similarity score ($S_{\text{BRE}}$), which avoids inflation from superhuman scores. For a normalized score $s_i$ on metric $i$, we compute $\text{BRE}_i = |s_i - 1| / \max(1, s_i)$ and aggregate as $S_{\text{BRE}} = 1 - \frac{1}{n} \sum_{i=1}^{n} \text{BRE}_i$. Thus, $\text{BRE}_i$ remains bounded in $[0, 1]$ even if $s_i > 1$.

Overall, we find competitive alignment across models, with MICRO-LLAMA-1B showing superior alignment compared to its counterparts. Panel (a) reports the average similarity score ($S_{\text{BRE}}$) aggregated across the 10 behavioral metrics for both MICRO-LLAMA-{1B, 3B} models, while panel (b) breaks down the human-normalized scores for each metric separately across the three post-trained models for the LLAMA-3.2-1B base model. Finally, panel (c) illustrates input examples from two of the seven classical psychological experiments included in COGBENCH, which are verbalized for LLM evaluation following the original benchmark design. More results in Appendix I.

## 5.5 COMPETITIVE PERFORMANCE ON REASONING BENCHMARKS

Here, we test whether our MICRO models incur any performance degradation relative to their two baselines. Figure 7 shows performance on GSM8K, MINERVA-MATH, MMLU, MMLU-PRO, and BBH, along with their average. Models are evaluated using fewshot chain-of-thought prompting, except for GSM8K, which is evaluated under zero-shot CoT prompting. For both base models, MICRO matches or exceeds comparable MoB baselines, while ablating the least relevant expert (i.e., the social expert for these benchmarks) further improves performance. We conduct pairwise Welch's $t$-tests between models and report significance directly in the plot. Results show that some base models, such as LLAMA-3.2-1B, benefit significantly from brain-like specialization, whereas others, such as LLAMA-3.2-3B, only show significant differences relative to their baselines on some benchmarks. We report additional results for the other models and benchmarks in Appendix F. We further show that our method is robust to different post-training pipelines, including DPO (Rafailov et al., 2023) and domain-specific instruction tuning (Appendix G).

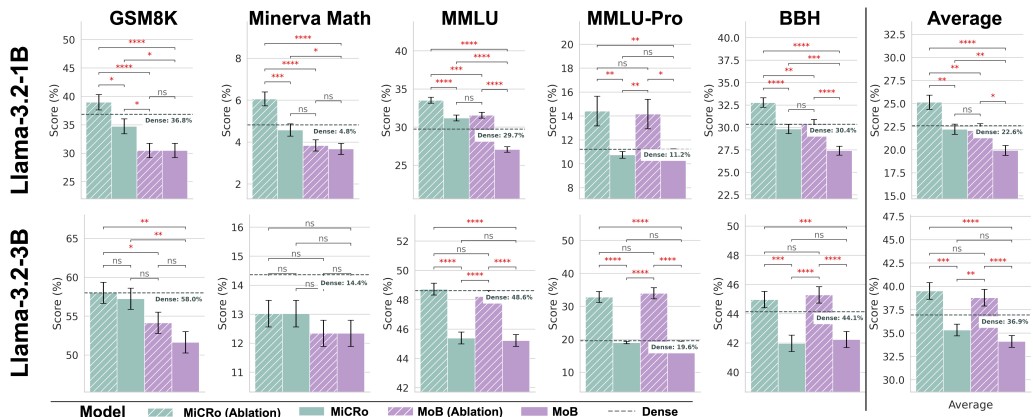

Figure 7: **Competitive Performance.** Results on GSM8K (0-shot CoT), MINERVA-MATH, MMLU, and BBH (fewshot CoT). MICRO matches or outperforms baselines, and ablating the least relevant expert (e.g., social for math benchmarks) yields further gains. For MoB (ABLATION) and MICRO (ABLATION) (on MMLU and BBH subtasks), results reflect the best performance obtained when ablating up to one expert. Significance is assessed with pairwise Welch's $t$-tests (shown in plot; $*$ denotes significant difference, while n.s. means not-significant). The dense model is shown as a dashed line. Results of the remaining models and on more benchmarks are provided in Appendix F.

## 6 DISCUSSION & FUTURE WORK

**Extending Specialization Beyond Cognitive Domains**   While inspired by the brain's functional organization, our specialization framework can be applied to any meaningful partition of expertise, such as technical domains or natural languages. One key question is whether the model preserves the intended specialization through the large-scale end-to-end training. Our results suggest that brain-inspired partitions provide a robust inductive bias; they persist throughout training and lead to structured, interpretable routing patterns. Supporting evidence in Appendix J shows that expert usage remains consistent across checkpoints over the course of Stage 3 training. Looking ahead, this framework could also be extended to other cognitive domains. For example, recent neuroscience findings point to a distinct brain network involved in abstract formal reasoning such as induction and deduction and another network for intuitive physics (Kean et al., 2025a;b). Incorporating a corresponding module could improve the model's performance on tasks involving such capacities.

**The Crucial Role of Stage-1 Pretraining Data**   Our experiments highlight the importance of the curated MICRO_SFT dataset in inducing effective specialization. Notably, we used only 3,055 samples in Stage-1, suggesting that even minimal domain-aligned supervision can shape expert behavior. This finding raises the possibility that different or more expansive data mixtures could further strengthen functional specialization and lead to additional gains in the model's behavior.

**Towards Brain Alignment Beyond Language**   Since our model is explicitly designed to mirror distinct cognitive networks in the human brain, and given that established neuroscience localizers can identify the corresponding expert modules, an exciting direction for future work is to examine whether the internal representations of these experts align more closely with neural activity in their respective brain networks (Schrimpf et al., 2018; 2020). Prior studies have shown that language-selective units in large language models correlate more strongly with activity in the human language network than randomly selected units (AlKhamissi et al., 2025a), suggesting a meaningful link between specialization in models and brains. This raises the natural question of whether similar alignment can be observed for other cognitive domains, such as reasoning or social cognition. However, assessing MICRO's neural alignment beyond the language network is currently limited by the lack of suitable datasets. Existing fMRI benchmarks rarely engage non-language regions such as the Multiple Demand (Duncan, 2010) network and often use blocked designs that preclude item-level analyses—highlighting the need for experimentalists to collect new datasets that explic-

itly target non-language brain regions. We believe that once suitable neural datasets exist, our model can be used to instantiate specific hypotheses about how these networks—and their corresponding experts—interact and exchange information.

**Limitations**  While our approach improves interpretability without sacrificing performance, several open questions remain. Scaling beyond an 8B base model has yet to be demonstrated, and the impact of adding more experts to the current MɪCRO architecture is still unknown. The MɪCRO$_{SFT}$ dataset used in Stage-1 ($\approx$ 3,000 GPT-4o pseudo-labeled samples) has not been evaluated for size sensitivity, leaving open whether increasing or reducing its scale would alter the degree of specialization or downstream performance. Although GPT-4O provides high-quality pseudo-labels, as demonstrated in Appendix B, using human-annotated data could strengthen the inductive bias and potentially improve the final model. We expect, however, that the potential of this approach will continue to grow as synthetic labellers become more accurate and reliable, enabling even stronger and more scalable specialization in future versions of this class of methods. Further, our post-training pipeline currently includes only SFT and DPO (Appendix G); exploring additional stages such as RLVR remains an avenue for future work. Finally, our evaluation of alignment to human behavior focuses on CogBench, and extending this analysis to a broader set of behavioral or cognitive datasets is an important direction for future research.

## 7 CONCLUSION

This work presents the *Mixture of Cognitive Reasoners* (MɪCRO), a modular architecture and training paradigm inspired by the functional specialization of the human brain. By aligning expert modules with distinct cognitive domains—language, logic, social reasoning, and world modeling—each reflecting the functionality of well-studied brain networks, we show that incorporating cognitive inductive biases into transformer models can effectively promote functional specialization. This results in improved behavioral alignment as measured by CoGBENCH, enhanced interpretability, all while being competitive or outperforming comparable models on reasoning benchmarks. Our staged training approach leverages a small curated dataset and enables specialization to emerge in a controllable and data-efficient manner. Furthermore, we show that the resulting modularity allows for targeted interventions at inference time, enabling the model to favor one mode of reasoning over another. These findings highlight a promising path toward more transparent, steerable, and cognitively grounded language models.

## ETHICS STATEMENT

This work involves two forms of human-derived data. First, all annotations newly introduced in this paper (Appendix B) were performed exclusively by the authors; no external annotators or study participants were recruited, and no personal or sensitive data were collected. Accordingly, this does not constitute human-subject research under typical institutional guidelines, and formal ethics approval (e.g., IRB review) was not required. Second, we use a public dataset of behavioral annotations (Tuckute et al., 2024b) in Appendix D, which was collected and released with approval from the corresponding university's Institutional Review Board.

## ACKNOWLEDGMENTS

We thank the members of the EPFL NLP and NeuroAI labs for their valuable feedback and insightful suggestions. We also gratefully acknowledge the support of the Swiss National Science Foundation (No. 215390), Innosuisse (PFFS-21-29), the EPFL Center for Imaging, Sony Group Corporation, and a Meta LLM Evaluation Research Grant. G.T. acknowledges support from MIT's McGovern Institute for Brain Research. This work has been made possible in part by a gift from the Chan Zuckerberg Initiative Foundation to establish the Kempner Institute at Harvard University.

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

APPENDIX

## A  CONSTRUCTING THE EXPERTS DATASET

**Datasets**   To construct the small curated datasets used in stages 1 and 2 of our training curriculum, we first identified existing datasets that align with the cognitive domain of each expert. Table 1 lists these datasets, the number of examples sampled from each, the corresponding high-level cognitive skill they were chosen to represent, and whether we used O1 to generate responses or relied on the original reasoning chains provided with the dataset.

Table 1: **Datasets Used to Induce Specialization in Stage-1.** Overview of datasets used to induce expert specialization during stages 1 and 2. Each dataset is aligned with a cognitive skill targeted by a specific expert. We indicate the number of examples sampled from each dataset and whether responses were generated using O1 or taken directly from the dataset's original reasoning chains.

| Expert | Task | Dataset | # Samples | Use O1 |
|---|---|---|---|---|
| Logic | Math | O1-Journey (Qin et al., 2024) | 327 | No |
| | | Math (Li et al., 2023) | 200 | No |
| | | GSM8K (Cobbe et al., 2021) | 100 | Yes |
| | Logic | Folio (Han et al., 2022) | 100 | Yes |
| | | LogicQA (Liu et al., 2020) | 100 | Yes |
| | Physics | Physics (Li et al., 2023) | 200 | No |
| **Total (Logic)** | | | **1027** | |
| Social | Pragmatics | Deceits (Hu et al., 2023) | 20 | Yes |
| | | Indirect Speech (Hu et al., 2023) | 20 | Yes |
| | | Irony (Hu et al., 2023) | 25 | Yes |
| | | Maxims (Hu et al., 2023) | 19 | Yes |
| | | Metaphor (Hu et al., 2023) | 20 | Yes |
| | | Humor (Hu et al., 2023) | 25 | Yes |
| | | Coherence (Hu et al., 2023) | 40 | Yes |
| | Emotion Detection | EmoCause (Kim et al., 2021) | 100 | Yes |
| | Theory of Mind | FanToM 1st Order (Kim et al., 2023) | 100 | Yes |
| | | FanToM 2nd Order (Kim et al., 2023) | 100 | Yes |
| | | BigToM (Gandhi et al., 2023) | 128 | Yes |
| | Social Reasoning | Mixture ProlificAI/social-reasoning-rlhf | 531 | Yes |
| **Total (Social)** | | | **1028** | |
| World | World Knowledge | Biology (Li et al., 2023) | 100 | No |
| | | Chemistry (Li et al., 2023) | 100 | No |
| | | PIQA (Bisk et al., 2020) | 200 | Yes |
| | | WikiQA (Yang et al., 2015) | 100 | Yes |
| | Spatial Reasoning | SpatialEval (Wang et al., 2024) | 100 | Yes |
| | Temporal Reasoning | TextTemporal (Li et al., 2025) | 100 | Yes |
| | World Building | World Building archit11/worldbuilding | 200 | No |
| | Cause Effect | CoPA (Wang et al., 2022) | 100 | Yes |
| **Total (World)** | | | **1000** | |

**Generating Reasoning Responses**   Since most of the existing datasets we identified (listed in Table 1) do not include reasoning steps for the final answer, we used O1 to generate them. Specifically, we prompted the model with the input followed by "Let's think step by step." to elicit a longer response that includes intermediate reasoning before reaching the final answer. This was only done for datasets that did not already contain suitable reasoning chains.

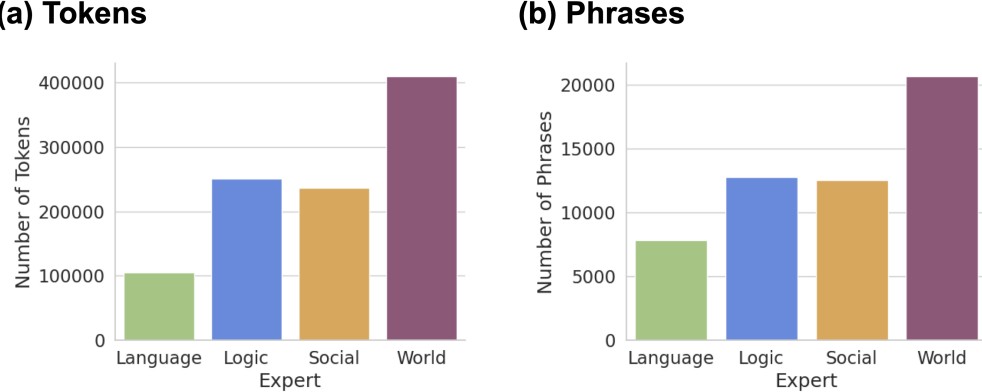

Figure 8: **Distribution of Expert Assignments across Tokens and Phrases. (a)** The distribution of expert assignments across tokens using the LLAMA-3.2-1B tokenizer. **(b)** The distribution of expert assignments labeled using GPT-4O for each phrase in the provided response.

**Pseudo-Labeling Responses** Finally, once we obtained responses with intermediate reasoning steps for all sampled examples, we used GPT-4O to pseudo-label each phrase in the response. During stage-1 training, each token in a phrase was then assigned to the expert identified by the pseudo-label. This labeling process was guided by the prompt shown in Figure 9. Figure 11 provides examples of labeled responses, while Figure 8 shows the distribution of expert assignments across tokens and phrases.

## B    INTER-ANNOTATOR AGREEMENT ANALYSIS OF MICRO$_{\text{SFT}}$

We assess the reliability of the MICRO$_{\text{SFT}}$ pseudo-labels using a set of standard agreement metrics between three human annotators and the GPT-4O labels on a subset of the MICRO$_{\text{SFT}}$ dataset. For the three human annotators, we report Krippendorff's $\alpha$ (a chance-corrected multi-annotator reliability metric), Fleiss' $\kappa$ (multi-rater extension of Cohen's $\kappa$), and pairwise Cohen's $\kappa$ with percent agreement. We also measure the agreement between the humans and the LLM labels both against the human majority vote and by treating the LLM as a fourth annotator.

### B.1    HUMAN–HUMAN AGREEMENT

Table 2 summarizes agreement across the three human annotators. Both Krippendorff's $\alpha$ and Fleiss' $\kappa$ indicate moderate reliability (0.517). The three-way exact agreement rate (all annotators assign the same label) is 49.4%.

Pairwise metrics (Table 3) reveal that annotators H1 and H2 exhibit stronger mutual agreement ($\kappa = 0.681$) relative to H1–H3 and H2–H3, which show lower yet nontrivial agreement levels ($\kappa \approx 0.43$–0.45).

Table 2: Agreement among human annotators.

| Metric | Value |
|---|---|
| Krippendorff's $\alpha$ (nominal, 3 annotators) | 0.517 |
| Fleiss' $\kappa$ (3 annotators) | 0.517 |
| 3-way exact agreement | 49.4% |

### B.2    HUMAN–LLM AGREEMENT

We evaluate human–LLM agreement in two ways: (1) comparing the LLM to the majority-vote label of the three humans, and (2) treating the LLM as a fourth annotator.

**GPT-4o Prompt for Pseudo-labeling O1 Responses**

I am training a mixture-of-experts (MoE) model that routes tokens individually (token-level routing) to specialized experts. The model includes four distinct experts, each clearly analogous to a specific cognitive network in the human brain. Below is a detailed explanation of each expert, along with examples of the types of tasks or token sequences each should typically handle:

- **Language Network (LN)**: Primarily responsible for linguistic processing, grammatical structures, vocabulary usage, syntax, semantics, and sentence coherence. This expert should handle tasks involving language comprehension, text fluency, sentence construction, paraphrasing, and interpreting linguistic nuances.
    - Example tasks: Completing sentences, grammar correction, paraphrasing sentences, translating between languages, summarizing text.

- **Multiple Demand Network (MD)**: Specializes in analytical thinking, mathematical calculations, numerical reasoning, and logical problem-solving. This expert engages explicitly in arithmetic operations, logical deductions, comparisons, quantitative reasoning, and systematic analysis.
    - Example tasks: Performing arithmetic operations, solving logical puzzles, analyzing numerical data, interpreting mathematical expressions, evaluating logical arguments.

- **Theory of Mind Network (ToM)**: Dedicated to social cognition and interpersonal reasoning. This expert interprets and predicts social interactions, emotional states, intentions, desires, beliefs, and motivations of individuals or groups.
    - Example tasks: Inferring a person's feelings or intentions from their actions, understanding dialogue involving interpersonal relations, predicting characters' behaviors based on emotional context, interpreting subtle social cues.

- **Default Mode Network (DMN)**: Responsible for integrating general world knowledge, context understanding, background information retrieval, and conceptual reasoning about everyday scenarios or common-sense understanding.
    - Example tasks: Providing background knowledge on common scenarios, contextualizing real-world situations, recalling general facts, understanding cause-and-effect relationships, providing narrative context.

Given the following example of model-generated text in response to a prompt, carefully label each token sequence with the expert best suited to handle it (LN, MD, ToM, or DMN). Ensure each sequence is assigned to only one expert. Output your answer clearly and explicitly in the following JSON format with each {sequence_i} corresponding to the actual sequence of tokens:

```json
{
  "{sequence_1}": "{expert_label}",
  "{sequence_2}": "{expert_label}",
  "{sequence_3}": "{expert_label}",
  ...
}
```

## Prompt

{prompt}

## Generation

{generation}

Figure 9: **Prompt Used for Pseudo-Labeling O1 Responses** The prompt used to instruct GPT-4O to label the O1 model generations given a specific input prompt.

**GPT-5 Prompt for Generating Expert Specific Stimuli**

{BRAIN_NETWORKS_DESCRIPTION}

Using the description above and your knowledge of the mentioned brain networks, generate **{N}** prompt and chain of thought answer pairs that activate only the **{BRAIN_NETWORK}** expert and do not activate any of the other experts. Output the results only and nothing else in JSONL format. Each JSON object should be in an LLM chat format as the following.

```
{"user": <prompt>, "assistant": <cot-answer>}
```

Figure 10: **Expert-Specific Prompt Template Used with GPT-5** Prompt provided to GPT-5 for generating expert-specific question–answer pairs. The stimuli for each expert was prompted separately using the same brain-network descriptions as in Figure 9.

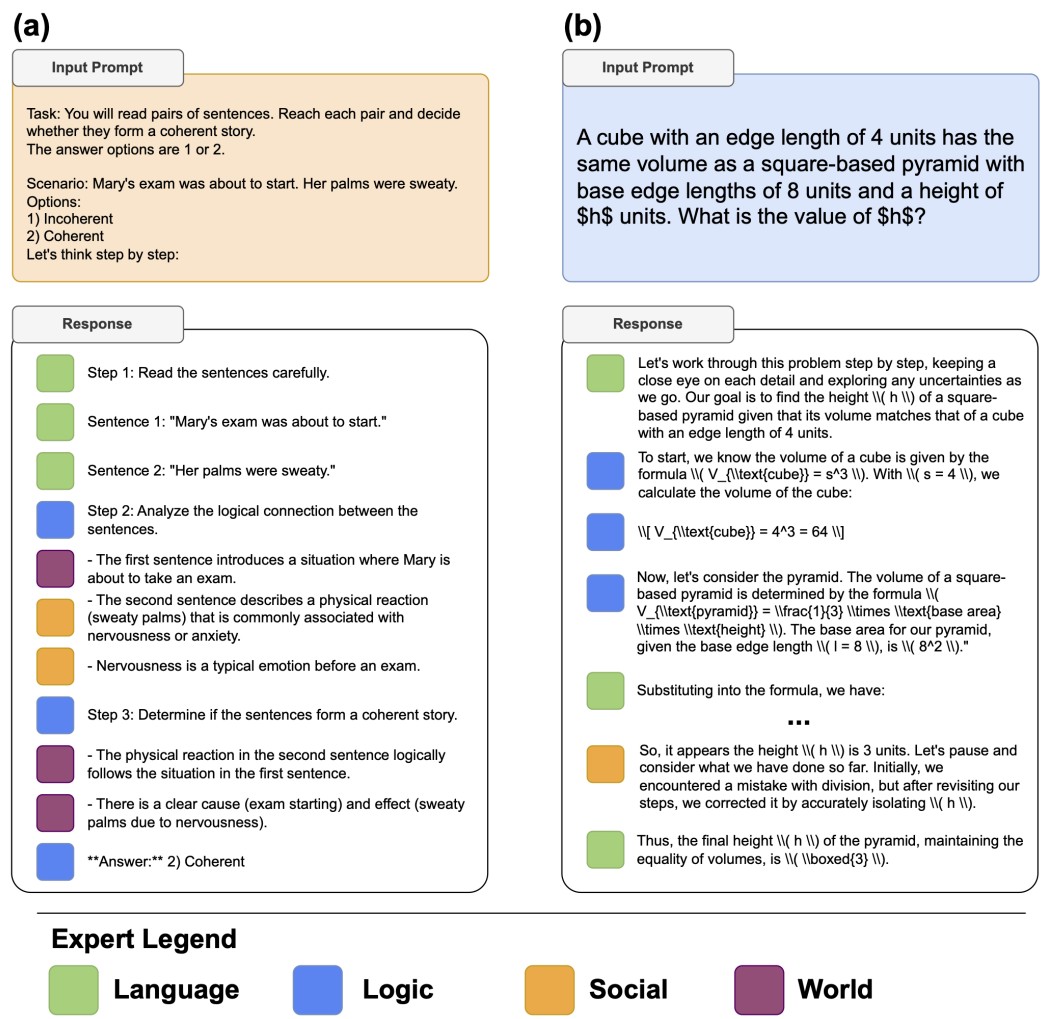

Figure 11: **Examples of Pseudo-Labeled Responses using GPT-4O (a)** shows a response generated by O1 for a prompt from the coherence subset of the PRAGMATICS dataset (Hu et al., 2023). **(b)** shows a response taken directly from the O1-JOURNEY dataset (Qin et al., 2024). Each subfigure includes the original prompt, the full model-generated response, and the corresponding pseudo-labels assigned to each phrase.

Table 3: Pairwise agreement between human annotators.

| Annotator Pair | Percent Agreement | Cohen's $\kappa$ |
|---|---|---|
| H1–H2 | 76.3% | 0.681 |
| H1–H3 | 58.9% | 0.431 |
| H2–H3 | 58.5% | 0.446 |

**LLM vs. Human Majority Vote.** Out of the full set, 240 items had a unique human majority label (13 items exhibited three-way ties). On this subset, the LLM achieves the performance shown in Table 4. The Cohen's $\kappa$ between the LLM and the majority vote is $0.533$, indicating substantial agreement comparable to human–human levels.

Table 4: LLM agreement with human majority-vote labels.

| Metric | Value |
|---|---|
| Accuracy | 0.658 |
| Macro F1 | 0.666 |
| Cohen's $\kappa$ (LLM vs. majority) | 0.533 |

**LLM as a Fourth Annotator.** We also compute multi-annotator reliability including the LLM (Table 5). Krippendorff's $\alpha$ decreases slightly to $0.497$, reflecting the LLM's moderate alignment with the human annotators.

Pairwise comparisons between each human annotator and the LLM (Table 6) show agreement levels similar to those between some human pairs (particularly H1–H3 and H2–H3).

Table 5: Agreement among 3 humans + LLM.

| Metric | Value |
|---|---|
| Krippendorff's $\alpha$ (nominal, 4 annotators) | 0.497 |

Table 6: Pairwise agreement between humans and the LLM.

| Annotator Pair | Percent Agreement | Cohen's $\kappa$ |
|---|---|---|
| H1–LLM | 62.8% | 0.489 |
| H2–LLM | 62.5% | 0.492 |
| H3–LLM | 59.7% | 0.448 |

Overall, the dataset exhibits moderate inter-annotator consistency across all metrics, with variation typical of multi-class subjective labeling tasks. The LLM aligns with human labels at levels comparable to human–human agreement, and the LLM agrees more with H1 and H2, who have a higher inter-annotator agreement.

## C  TOKEN ROUTING PATTERNS

**GPT-5 Prompt for Generating Expert Specific Stimuli**  Figure 10 presents the prompt used to instruct GPT-5 to generate the question–answer pairs shown in the non-benchmark token-routing pattern plots. The descriptions of the brain networks are identical to those in Figure 9, which were previously used to pseudo-label O1-generated responses for constructing the MICRO$_{\text{SFT}}$ dataset. We queried GPT-5 separately for each expert. We show the routing patterns for additional models in Figure 12 and for MOE models in Figure 17.

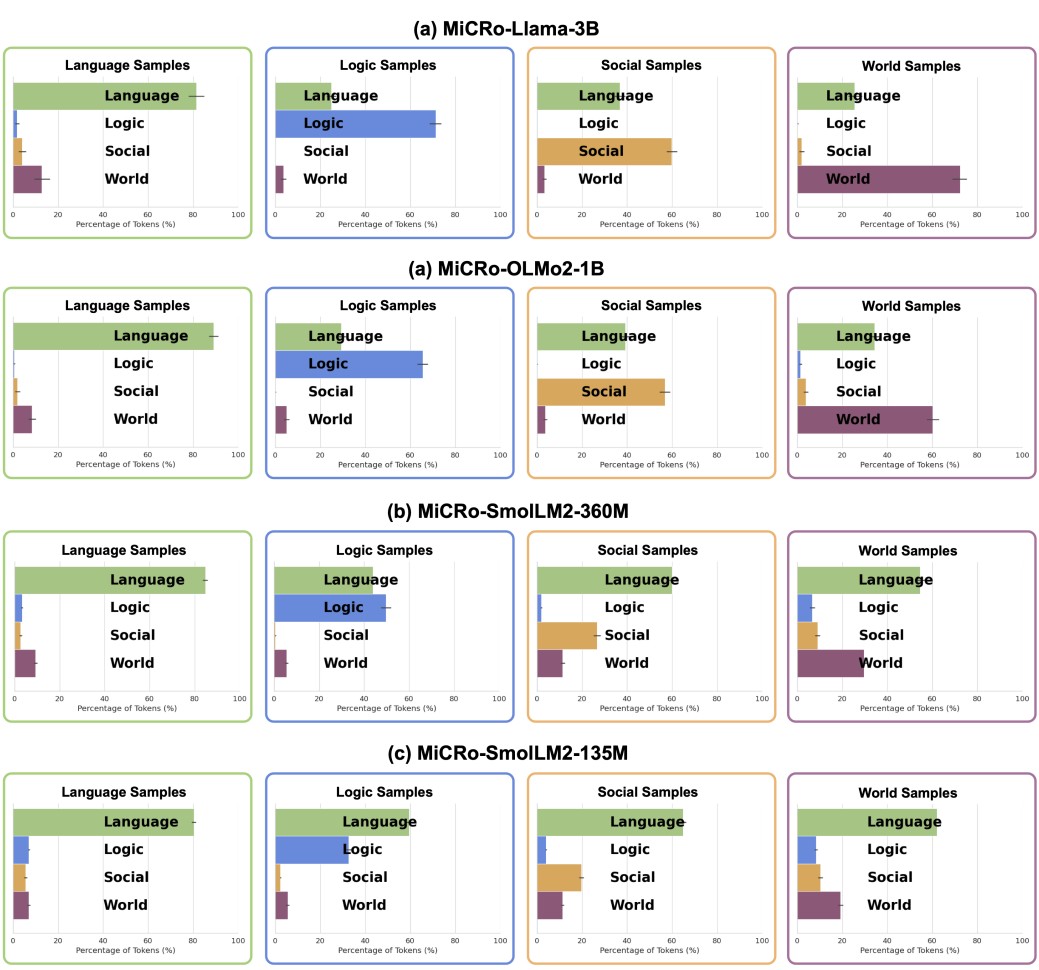

Figure 12: **Token Routing Patterns in Additional MICRO-Models.** Percentage of tokens routed to each expert, aggregated across all layers, for additional MICRO models. Distributions are computed over GPT-5–generated question–answer pairs designed to engage specific domains. Results show consistent brain-inspired specialization, with tokens preferentially assigned to the relevant experts depending on the task domain. Figure 13 shows the corresponding layer-wise token routing of these plots, while Figure 14 shows the token routing on benchmark testing data.

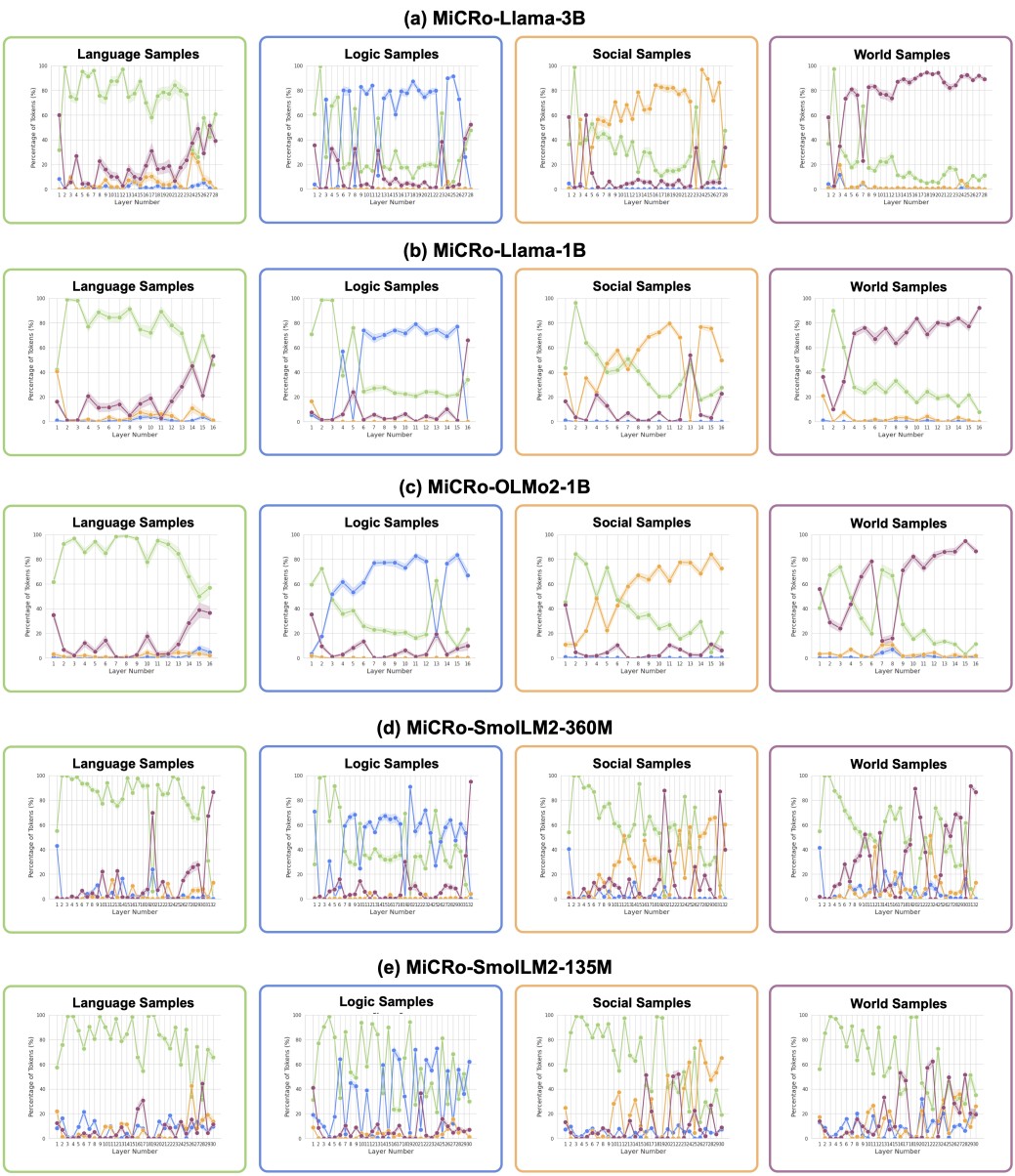

Figure 13: **Layer-wise Token Routing in MICRO Models**. Token routing distributions across layers for five MICRO models, measured on GPT-5–generated question–answer pairs targeting specific domains. In all models, the language expert is consistently engaged in early layers, while domain-specific experts (logic, social, world) are increasingly activated in deeper layers. This hierarchical organization parallels findings from cognitive neuroscience, where linguistic processing precedes engagement of higher-level networks.

**Layerwise Routing Patterns** Figure 13 illustrates layer-wise token routing patterns for five MI-CRo models. Surprisingly, consistent trend emerges: tokens are initially processed by the language expert before being delegated to higher-level experts depending on the task domain. This organization parallels findings in cognitive neuroscience, where the language network is engaged early for virtually all linguistic input and then interfaces with other specialized networks (such as multiple-demand or social cognition systems) depending on task demands (Fedorenko et al., 2024). In Figure 14, we show benchmark-specific token routing patterns across layers as well. To probe social specialization directly, we also include evaluation on the EMPATHY benchmark (Buechel et al., 2018), which primarily engages the social expert, further confirming the expected routing behavior is generalizable across datasets.

## D  CORRELATION WITH HUMAN JUDGMENTS

We use a dataset of 1,000 six-word sentences from Tuckute et al. (2024b), each annotated with human ratings across several behavioral dimensions, collected independently of our routing framework. To test correlations with human judgments, we selected features expected to align with specific experts: GRAMMATICALITY and PLAUSIBILITY with the language expert, MENTAL STATES with the social expert, and PHYSICAL OBJECTS and PLACES with the world expert. The dataset does not include features relevant to the logic expert.

To analyze these relationships, we divide each model into three layer segments (early, middle, late) and averaged router probabilities within each segment. Figure 15 reports correlations between the average routing probability of each expert and human ratings for MICRO-LLAMA-3.2-1B and MICRO-LLAMA-3.2-3B. For both models and layer segments, mental state ratings correlate most strongly with the social expert. Language expert probabilities correlate with GRAMMATICALITY and PLAUSIBILITY, but primarily in early layers. PHYSICAL OBJECTS and PLACES correlate with the world expert, while the logic expert shows no positive correlations (and in most cases negative correlations) with these features. These findings suggest that our router exhibits a meaningful degree of correspondence with human behavioral judgments.

## E  ADDITIONAL EXPERT ABLATION RESULTS

Figure 16 reports the effect of ablating individual experts, including the language expert, on benchmark performance for five MICRO models. We find that the language expert is essential for most tasks, while domain-specific experts—such as the logic expert for GSM8K and MINERVA MATH—are also necessary to maintain performance. Interestingly, in some cases, ablating an expert improves performance, suggesting that certain experts may interfere with more relevant ones, leading to performance degradation when all are active.

## F  BENCHMARKS

Table 7: **Number of Shots and Samples Per Benchmark Used in Evaluation.** Number of shots and samples used when evaluating the test-set of each benchmark. Last two row shows whether we used CoT or evaluated using log-probabilities and the metric used to obtain the final accuracy.

| Benchmark | GSM8K | Minerva Math | MMLU | MMLU-Pro | BBH | HellaSwag | PIQA | ARC$_{Easy}$ | ARC$_{Challenge}$ |
|---|---|---|---|---|---|---|---|---|---|
| N-Shots | 0-Shot | 4-shots | 4-shots | 5-shots | 3-Shots | 0-Shot | 0-Shot | 0-Shot | 0-Shot |
| Num Samples | 1,319 | 5,000 | 14,042 | 12,032 | 6,511 | 10,042 | 1,838 | 2,376 | 1,172 |
| CoT Prompting | Yes | Yes | Yes | Yes | Yes | No | No | No | No |
| Metric | Exact Match | Exact Match | Exact Match | Exact Match | Exact Match | Acc Norm | Acc Norm | Acc | Acc Norm |

**Benchmarks Description** We evaluate our models on eight benchmarks using various fewshot settings, four of which are prompted to generate a reasoning chain before producing the final answer. These reasoning steps are intended to more meaningfully engage the expert modules throughout the generation process, which is why we focused on them in the main paper. The other benchmarks are multiple choice questions where the most likely candidate—as measured by the log-probabilities of the model—is taken as the prediction. Table 7 lists the number of in-context examples used

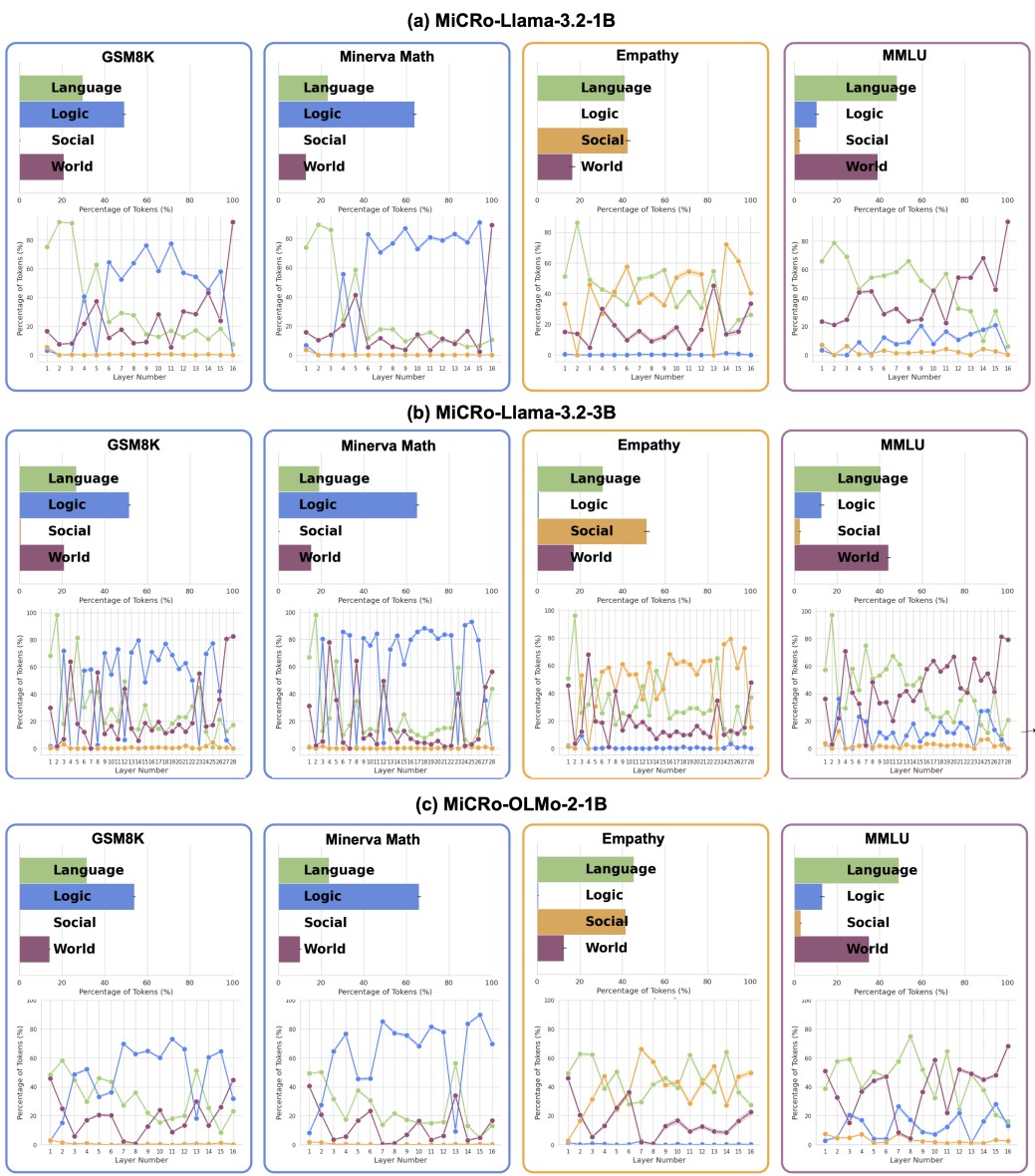

Figure 14: **Benchmark Token Routing Patterns.** Token routing patterns for **(a)** MICRO-LLAMA-3.2-1B, **(b)** MICRO-LLAMA-3.2-3B, and **(c)** MICRO-OLMO-2-1B, evaluated on up to 1,000 samples drawn from the GSM8K, MINERVA-MATH, EMPATHY, and MMLU test sets. For each model, the top panel reports the overall percentage of tokens routed to each expert across the whole model (variance across samples), while the bottom panel shows layer-wise routing. The latter reveals an emergent hierarchy: earlier layers emphasize language grounding, whereas deeper layers increasingly delegate to domain-relevant experts.

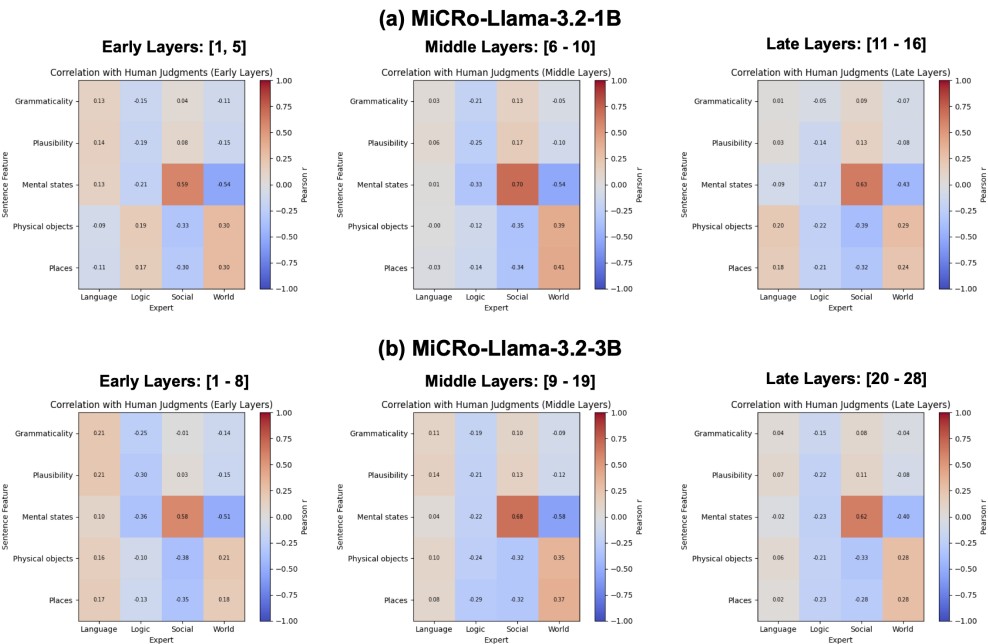

Figure 15: **Correlations Between Expert Routing Probabilities and Human Ratings.** Correlations are shown for MICRO-LLAMA-3.2-1B and MICRO-LLAMA-3.2-3B, averaged across early, middle, and late layer segments. Mental state ratings correlate most strongly with the social expert, grammaticality and plausibility correlate to some degree with the language expert (primarily in early layers), and physical objects and places with the world expert. The logic expert shows no positive correlations with these features.

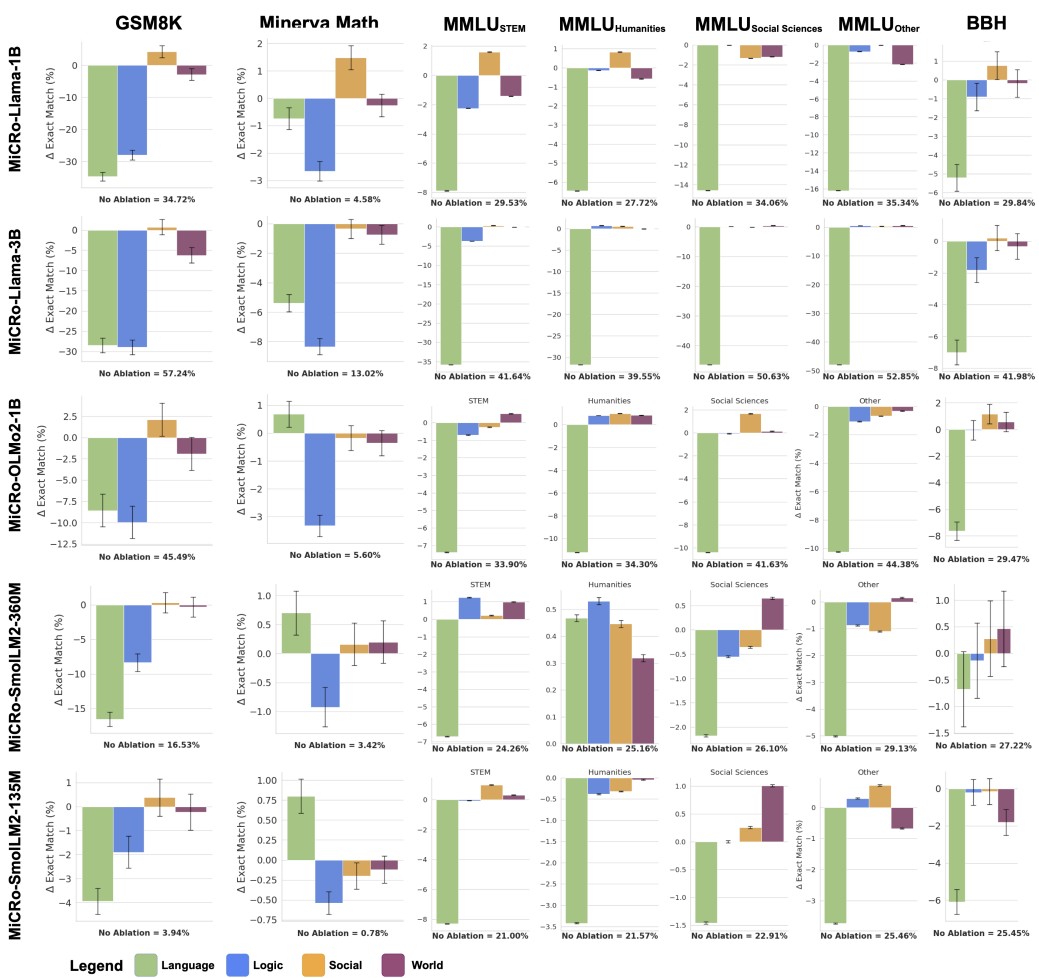

Figure 16: **Expert Ablation Results Across Benchmarks.** Impact of ablating individual experts on benchmark performance for five MICRO models. Results are shown for GSM8K, MINERVA MATH, BBH, and MMLU, with the latter divided into its four subcategories. Removing the language expert causes substantial drops across most tasks, while domain-specific experts (e.g., logic for math benchmarks) are critical for their respective domains. In some cases, ablating an expert improves performance, suggesting interference with more relevant experts.

Table 8: **Additional Benchmark Results for MiCRo and Baselines** Accuracy (%) ± standard error across reasoning and knowledge benchmarks. Results are reported for different model classes (Dense, MoB, and MiCRo) under each base model.

| Base Model | Model | GSM8K | Minerva | MMLU | MMLU$_{Pro}$ | BBH | ARC$_{Easy}$ | ARC$_{Challenge}$ | HellaSwag | PIQA |
|---|---|---|---|---|---|---|---|---|---|---|
| SmollM2-135M | Dense | $2.7 \pm 0.4$ | $0.5 \pm 0.1$ | $21.5 \pm 0.3$ | $7.8 \pm 0.2$ | $24.1 \pm 0.5$ | $62.8 \pm 1.0$ | $29.6 \pm 1.3$ | $43.6 \pm 0.5$ | $67.7 \pm 1.1$ |
| | MoB | $3.0 \pm 0.5$ | $0.6 \pm 0.1$ | $21.9 \pm 0.3$ | $7.4 \pm 0.2$ | $23.5 \pm 0.5$ | $63.0 \pm 1.0$ | $29.9 \pm 1.3$ | $43.5 \pm 0.5$ | $67.8 \pm 1.1$ |
| | MiCRo | $3.9 \pm 0.5$ | $0.8 \pm 0.1$ | $22.5 \pm 0.4$ | $7.9 \pm 0.2$ | $25.4 \pm 0.5$ | $56.0 \pm 1.0$ | $27.6 \pm 1.3$ | $41.8 \pm 0.5$ | $67.5 \pm 1.1$ |
| SmollM2-360M | Dense | $15.0 \pm 1.0$ | $3.7 \pm 0.3$ | $26.4 \pm 0.4$ | $9.9 \pm 0.3$ | $27.3 \pm 0.5$ | $69.7 \pm 0.9$ | $37.5 \pm 1.4$ | $56.6 \pm 0.5$ | $71.4 \pm 1.1$ |
| | MoB | $17.4 \pm 1.0$ | $3.9 \pm 0.3$ | $26.8 \pm 0.4$ | $9.8 \pm 0.3$ | $27.7 \pm 0.5$ | $70.0 \pm 0.9$ | $37.1 \pm 1.4$ | $56.9 \pm 0.5$ | $72.0 \pm 1.0$ |
| | MiCRo | $16.5 \pm 1.0$ | $3.4 \pm 0.3$ | $26.0 \pm 0.4$ | $10.1 \pm 0.3$ | $27.2 \pm 0.5$ | $69.9 \pm 0.9$ | $38.2 \pm 1.4$ | $56.7 \pm 0.5$ | $71.7 \pm 1.1$ |
| Llama-3.2-1B | Dense | $36.8 \pm 1.3$ | $4.8 \pm 0.3$ | $29.7 \pm 0.4$ | $11.2 \pm 0.3$ | $30.4 \pm 0.5$ | $64.3 \pm 1.0$ | $33.7 \pm 1.4$ | $58.4 \pm 0.5$ | $73.8 \pm 1.0$ |
| | MoB | $30.5 \pm 1.3$ | $3.7 \pm 0.3$ | $27.1 \pm 0.4$ | $11.0 \pm 0.3$ | $27.4 \pm 0.5$ | $61.7 \pm 1.0$ | $32.4 \pm 1.4$ | $56.2 \pm 0.5$ | $71.3 \pm 1.1$ |
| | MiCRo | $34.7 \pm 1.3$ | $4.6 \pm 0.3$ | $31.2 \pm 0.4$ | $10.7 \pm 0.3$ | $29.8 \pm 0.5$ | $59.6 \pm 1.0$ | $32.8 \pm 1.4$ | $54.7 \pm 0.5$ | $73.1 \pm 1.0$ |
| Llama-3.2-3B | Dense | $58.0 \pm 1.4$ | $14.4 \pm 0.5$ | $48.6 \pm 0.4$ | $19.6 \pm 0.4$ | $44.1 \pm 0.6$ | $73.6 \pm 0.9$ | $42.9 \pm 1.4$ | $68.9 \pm 0.5$ | $77.0 \pm 1.0$ |
| | MoB | $51.6 \pm 1.4$ | $12.3 \pm 0.5$ | $45.2 \pm 0.4$ | $19.1 \pm 0.4$ | $42.2 \pm 0.6$ | $71.6 \pm 0.9$ | $41.3 \pm 1.4$ | $67.3 \pm 0.5$ | $77.0 \pm 1.0$ |
| | MiCRo | $57.2 \pm 1.4$ | $13.0 \pm 0.5$ | $45.4 \pm 0.4$ | $19.0 \pm 0.4$ | $42.0 \pm 0.6$ | $73.0 \pm 0.9$ | $43.3 \pm 1.4$ | $67.4 \pm 0.5$ | $76.6 \pm 1.0$ |

and the number of samples tested for each benchmark. For the remaining benchmarks, we used the default fewshot examples from the `lm-evaluation-harness` (Gao et al., 2024) repository. Specifically, we used the `bbh_cot_fewshot` task for BBH, the `mmlu_flan_cot_fewshot` task for MMLU, the `mmlu_pro` for MMLU-PRO, the `minerva_math` task for MINERVA MATH, and the `gsm8k_cot_zeroshot` for the GSM8K task. We used the default tasks for the multiple-choice benchmarks.

**Extended Benchmark Results**  Table 8 reports results for additional base models as well as on benchmarks beyond those presented in the main paper. Consistent with the main results, MICRO remains comparable to Dense and MoB baselines across most tasks while being interpretable. These supplementary experiments provide further evidence that the observed trends hold across a broader range of model scales and evaluation settings.

# G  ROBUSTNESS ACROSS POST-TRAINING METHODS

We further assess the robustness of our method to different post-training methods by applying two variations. First, we further post-train our MICRO models using Direct Preference Optimization (DPO) (Rafailov et al., 2023) on a subset of the TÜLU-2.5 preference dataset (Ivison et al., 2024) (Table 9). Second, we replace the large-scale general-purpose TÜLU-3 dataset used in stage-3 with a more domain-specific (medical) instruction-tuning set used in Bosselut et al. (2024) (Table 10). Our results show that our method is robust to different post-training pipelines, whether applying DPO or using an alternative instruction-tuning dataset, as shown in Tables 9 and 10 respectively.

Table 9: **Performance After DPO Finetuning** Comparison of MICRO models and baselines after further finetuning with DPO on a preference dataset. Results show average performance across the 4 benchmarks, indicating that specialization remains beneficial after DPO.

| Base Model | Model | GSM8K | Minerva Math | MMLU | BBH | Average |
|---|---|---|---|---|---|---|
| Llama-3.2-1B | Dense | 38.1 | 3.9 | 29.4 | **30.3** | 25.4 |
| | MICRO | **39.3** | **5.8** | **31.8** | **30.3** | **26.8** |
| OLMo-2-1B | Dense | 45.8 | 5.6 | 39.3 | 29.8 | 30.1 |
| | MICRO | **48.1** | **5.8** | **39.8** | **30.4** | **31.0** |

# H  MIXTURE-OF-EXPERTS RESULTS

In the main paper, we report results using the mixture-of-blocks (MOB) architecture, where each expert is a full transformer block with its own attention mechanism. Here, we contrast these results with the more standard mixture-of-experts (MOE) architecture, where experts consist only of FFN blocks and attention is shared across experts within each layer. We first present routing patterns for

Table 10: **Performance on Medical Benchmarks After Domain-Specific Instruction Tuning.**
Models are finetuned during Stage-3 using a medical instruction-tuning dataset instead of TÜLU-3,
and evaluated on four medical benchmarks. Results show that specialization achieve competitive
performance across both base models, and outperforming in the out-of-distribution (OOD) setting.
We choose the option with the highest log-probability among the multiple-choice options.

| Base Model | Model | Out-of-Distribution | | In-Distribution | | |
| | | MMLU Medicine | MedQA | MedMCQA | PubMedQA | Average |
|---|---|---|---|---|---|---|
| Llama-3.2-1B | Dense | 26.0 | 34.3 | 33.9 | **73.4** | 41.9 |
| | MICRO | **28.3** | **35.2** | 33.9 | 71.4 | **42.2** |
| OLMo-2-1B | Dense | 35.8 | 34.2 | **35.3** | **74.0** | 44.8 |
| | MICRO | 35.8 | **36.3** | 34.5 | 73.8 | **45.1** |

Table 11: **Results with Mixture-of-Experts (MoE) Architectures.** Accuracy (%) ± standard error
is reported for Dense, MoE, and MICRO-MOE models across multiple benchmarks. For each base
model, the best score per benchmark is highlighted in bold.

| Base Model | Model | GSM8K | Minerva | MMLU | BBH | ARC$_{Easy}$ | ARC$_{Challenge}$ | HellaSwag | PIQA |
|---|---|---|---|---|---|---|---|---|---|
| SmollM2-135M | Dense | $2.7 \pm 0.4$ | $0.5 \pm 0.1$ | $21.5 \pm 0.3$ | $24.1 \pm 0.5$ | $62.8 \pm 1.0$ | $29.6 \pm 1.3$ | $43.6 \pm 0.5$ | $67.7 \pm 1.1$ |
| | MoE | $2.8 \pm 0.5$ | $0.6 \pm 0.1$ | $22.3 \pm 0.3$ | $24.6 \pm 0.5$ | $62.9 \pm 1.0$ | $29.4 \pm 1.3$ | $43.6 \pm 0.5$ | $67.6 \pm 1.1$ |
| | MiCRo-MoE | $4.1 \pm 0.5$ | $0.4 \pm 0.1$ | $22.2 \pm 0.3$ | $24.5 \pm 0.5$ | $62.0 \pm 1.0$ | $29.0 \pm 1.3$ | $43.4 \pm 0.5$ | $67.5 \pm 1.1$ |
| SmollM2-360M | Dense | $15.0 \pm 1.0$ | $3.7 \pm 0.3$ | $26.4 \pm 0.4$ | $27.3 \pm 0.5$ | $69.7 \pm 0.9$ | $37.5 \pm 1.4$ | $56.6 \pm 0.5$ | $71.4 \pm 1.1$ |
| | MoE | $16.1 \pm 1.0$ | $3.6 \pm 0.3$ | $26.6 \pm 0.4$ | $27.4 \pm 0.5$ | $70.2 \pm 0.9$ | $37.2 \pm 1.4$ | $56.8 \pm 0.5$ | $71.8 \pm 1.1$ |
| | MiCRo-MoE | $16.1 \pm 1.0$ | $4.0 \pm 0.3$ | $26.1 \pm 0.4$ | $27.0 \pm 0.5$ | $69.7 \pm 0.9$ | $37.5 \pm 1.4$ | $56.7 \pm 0.5$ | $71.4 \pm 1.1$ |
| Llama-3.2-1B | Dense | $36.8 \pm 1.3$ | $4.8 \pm 0.3$ | $29.7 \pm 0.4$ | $30.4 \pm 0.5$ | $64.3 \pm 1.0$ | $33.7 \pm 1.4$ | $58.4 \pm 0.5$ | $73.8 \pm 1.0$ |
| | MoE | $29.1 \pm 1.3$ | $4.7 \pm 0.3$ | $25.7 \pm 0.4$ | $28.6 \pm 0.5$ | $64.1 \pm 1.0$ | $35.2 \pm 1.4$ | $57.9 \pm 0.5$ | $72.5 \pm 1.0$ |
| | MiCRo-MoE | $35.4 \pm 1.3$ | $5.0 \pm 0.3$ | $30.4 \pm 0.4$ | $30.1 \pm 0.5$ | $65.0 \pm 1.0$ | $35.5 \pm 1.4$ | $57.3 \pm 0.5$ | $73.8 \pm 1.0$ |

MICRO-MoE models, highlighting cases where our training curriculum fails to induce the intended
specialization—an issue we primarily observe in models larger than 1.5B parameters. We then report
the performance of the models that did exhibit specialization on reasoning benchmarks.

## H.1 MOE TOKEN ROUTING PATTERNS

Figure 17 shows routing patterns for five MICRO-MoE models on question–answer pairs gener-
ated with GPT-5 to target specific experts. The MICRO-MOE-LLAMA-1B model exhibits the
intended specialization, whereas the 3B variant does not. Within the SMOLLM2 family, the 135M
and 360M models display partial specialization, though less cleanly than LLAMA-1B, often de-
faulting to the language expert regardless of the input domain. The 1.7B model fails to specialize,
similar to MICRO-MOE-LLAMA-3B, indicating that the MoE architecture does not reliably induce
the desired specialization under our training curriculum.

## H.2 MOE BENCHMARK RESULTS

Table 11 presents results for models trained with a Mixture-of-Experts (MoE) design, complemen-
tary to the Mixture-of-Blocks (MoB) results reported in the main paper. The key distinction between
MoE and MoB lies in what is replicated to form the experts. In standard MoE, only the feed-forward
network (FFN) within each layer is cloned into multiple experts, with the self-attention module
shared across all experts. In contrast, MoB duplicates the entire transformer block—including both
the attention and FFN components—so that each expert has its own attention mechanism as well as
its own FFN. We find that MoB scales more effectively: under our training curriculum, specializa-
tion emerges reliably in larger models ($>$,1B parameters) for MoB, but not for MoE. For this reason,
we focus on MoB in the main text and do not include the MoE variants of the other base models, as
they did not exhibit the expected functional specialization.

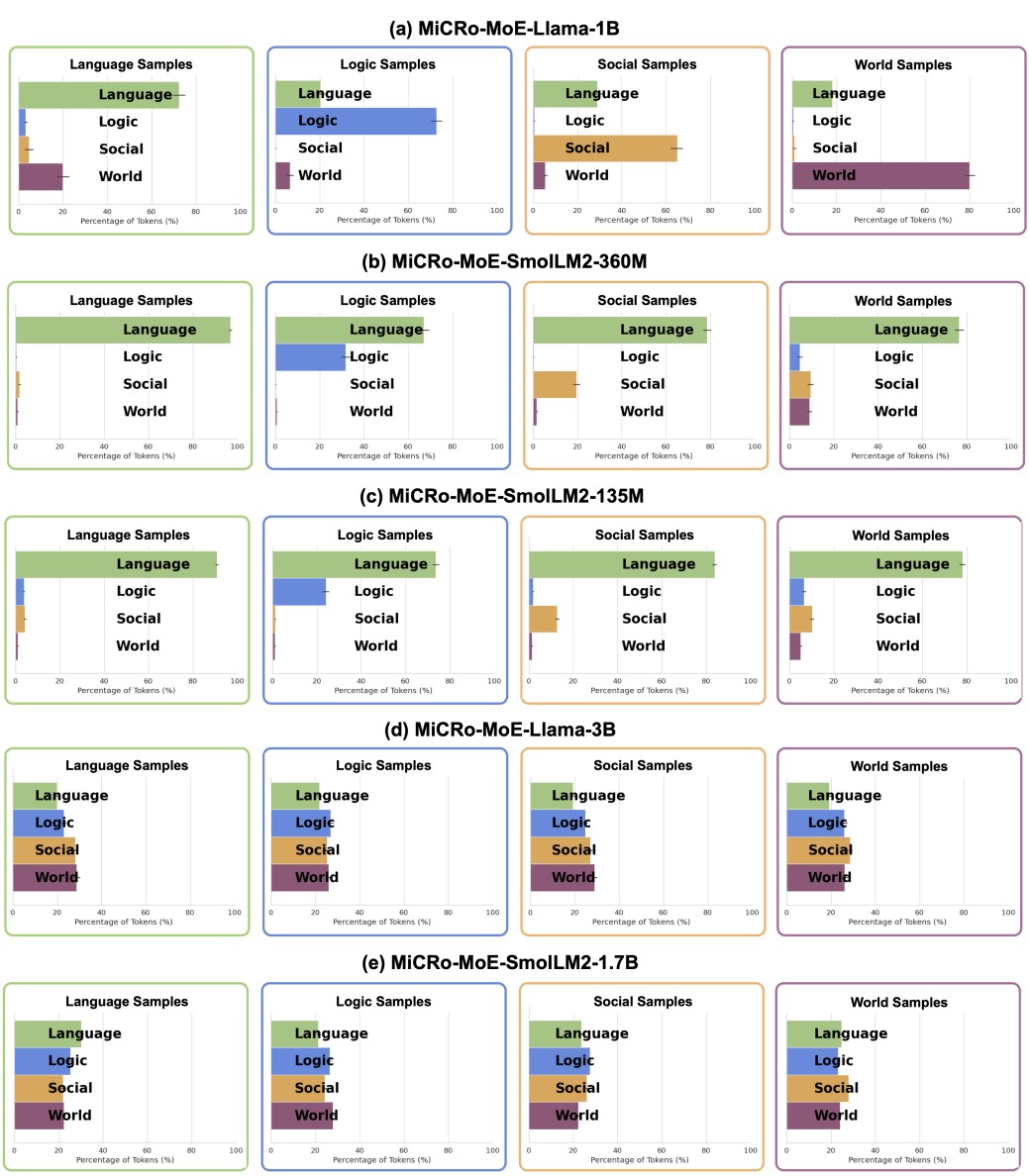

Figure 17: **Routing Patterns in MICRO-MoE Models.** Routing behavior for five MICRO-MoE models on GPT-5–generated question–answer pairs targeting specific experts. The MICRO-MOE-LLAMA-1B model shows the intended specialization, while larger variants (e.g., 3B, SmolLM2-1.7B) fail to specialize. Smaller SMOLLM2 models (135M and 360M) display partial but less consistent specialization, often defaulting to the language expert. These results suggest that the MoE architecture does not reliably induce brain-like specialization under our training curriculum.

Table 12: Performance of MICRO-OLMo-1B when increasing the number of active experts from Top-1 to Top-2 at test time. Enabling an additional expert leads to consistent improvements across most benchmarks, demonstrating that the model generalizes well to increased routing capacity even though SFT was performed with Top-1 routing. The benchmarks MATH, ARC-E, and ARC-C refer to MINERVA-MATH, ARC-EASY, and ARC-CHALLENGE respectively.

| K | GSM8K | BBH | MMLU | MATH | HellaSwag | PIQA | ARC-E | ARC-C | Avg |
|---|-------|-----|------|------|-----------|------|-------|-------|-----|
| 1 | 45.5% | 29.5% | 37.9% | 5.6% | 65.4% | 75.2% | 70.2% | 41.3% | 46.3% |
| 2 | **47.7%** | **30.6%** | **38.2%** | **6.8%** | **66.4%** | **75.4%** | **71.9%** | **41.6%** | **47.3%** |

## I  ADDITIONAL BEHAVIORAL ALIGNMENT RESULTS

Figure 18 shows alignment to human behavior for additional base models, comparing MICRO with corresponding MOB and DENSE baselines. We find that MICRO achieves higher average behavioral alignment on COGBENCH metrics in larger models, while maintaining comparable performance in smaller models. Please refer to §5.4 for more details on how we evaluate the models.

## J  SPECIALIZATION REMAINS CONSISTENT THROUGHOUT TRAINING

Figure 19 illustrates token routing assignments across checkpoints during Stage 3 training of MICRO-LLAMA-1B, with checkpoint-0 representing the final weights from Stage 2. The results show that the model consistently preserves the specialization established in stages 1 and 2, despite no explicit constraints being enforced during this phase, except for the initial weak inductive bias. This suggests that brain-like specialization may offer a robust initialization, enabling the model to maintain functionally distinct expert behaviors throughout continued end-to-end training.

## K  QUALITATIVE EXAMPLES OF STEERING BEHAVIOR

Figures 22-25 show examples of how one can use MICRO to steer the model's behavior by selectively ablating or activating certain experts. In the examples provided, we only retain the target expert along with the language expert using the MICRO models. When the social expert is ablated, the model shifts toward a more analytical tone, producing a response that is logically coherent but lacking in empathy.

## L  TEST-TIME SCALING BY INCREASING THE NUMBER OF ACTIVE EXPERTS

Table 12 reports the effect of increasing the number of active experts at test time from 1 to 2 for MICRO-OLMo-1B. The results show that test-time compute can be scaled by enabling additional experts, and that the model generalizes well to this setting and improves performance on all benchmarks, even though the large-scale SFT stage was trained exclusively with top-1 routing. This indicates that MICRO retains robustness when the routing capacity is expanded at inference time under the $k=2$ setting. However, with larger values of $k$ the performance degrades slightly.

## M  SCALING MICRO TO LLAMA-3.1-8B BASE MODEL

We post-trained three Llama-3.1-8B variants: (1) MICRO-LLAMA-8B, (2) its modular baseline LLAMA-8B-MOB, and (3) LLAMA-8B-DENSE. However, due to compute constraints, we instantiated experts only in the last 12 layers of MICRO-LLAMA-8B and LLAMA-8B-MOB, keeping the earlier layers dense. This choice is motivated by our prior findings (Figures 13–14), which show that early layers predominantly route to language experts, while non-language specializations emerge in later layers. The results confirm that MICRO-LLAMA-8B exhibits the expected routing patterns in its last 12 layers, as shown in Figure 20. Table 13 shows the results on 9 benchmarks of the three LLAMA-3.1-8B model variants, along with the results when we remove the most detrimental expert across all layers for a given task for the MICRO and MOB models. Using a paired Wilcoxon

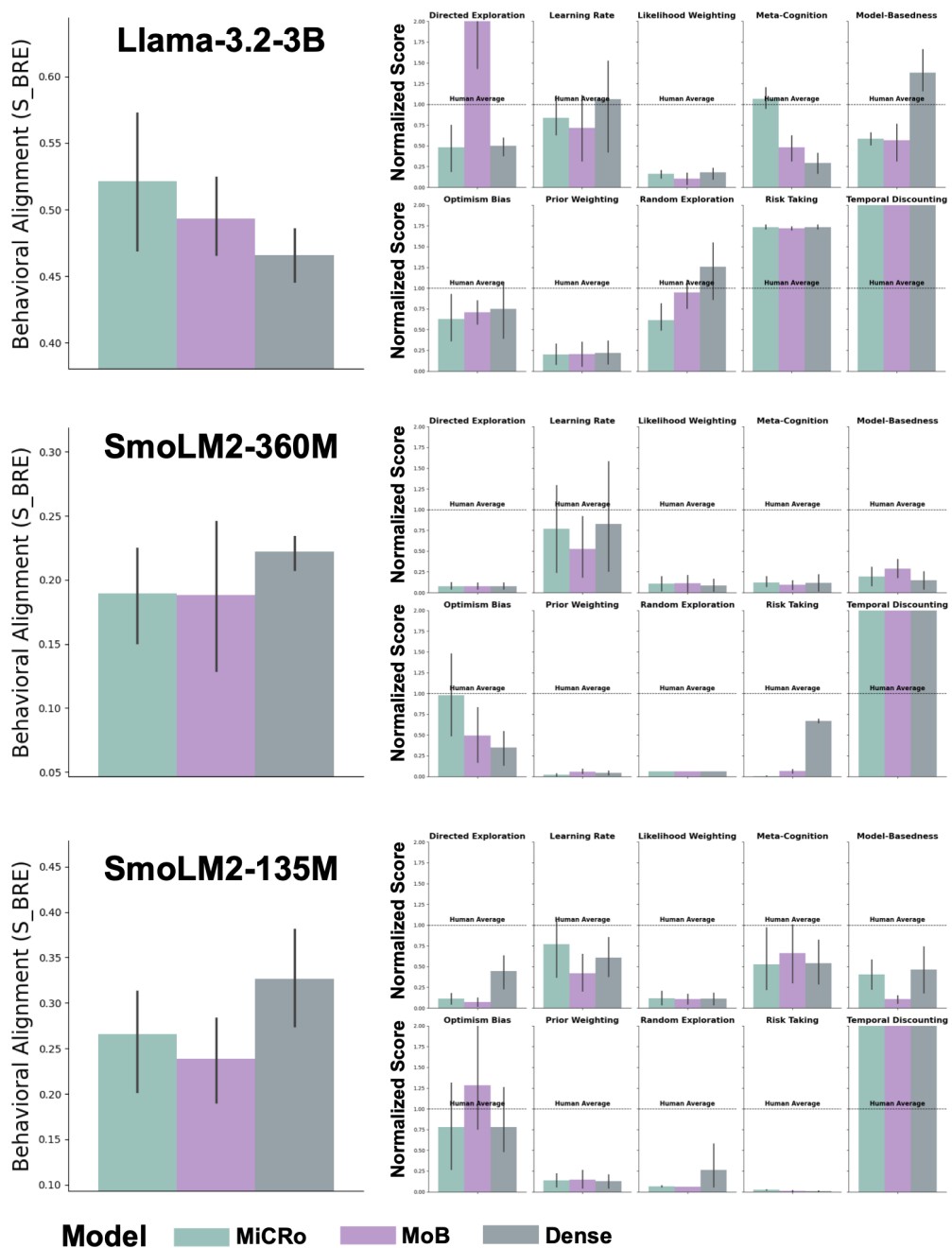

Figure 18: **Behavioral Alignment on Additional Base Models.** Results for LLAMA-3.2-3B, SMOLLM2-360M, and SMOLLM2-135M on COGBENCH. Left: average similarity to human behavior across all metrics. Right: fine-grained results for each behavioral metric. MICRO is compared with MOB and DENSE baselines, showing stronger alignment in larger models and comparable performance in smaller ones.

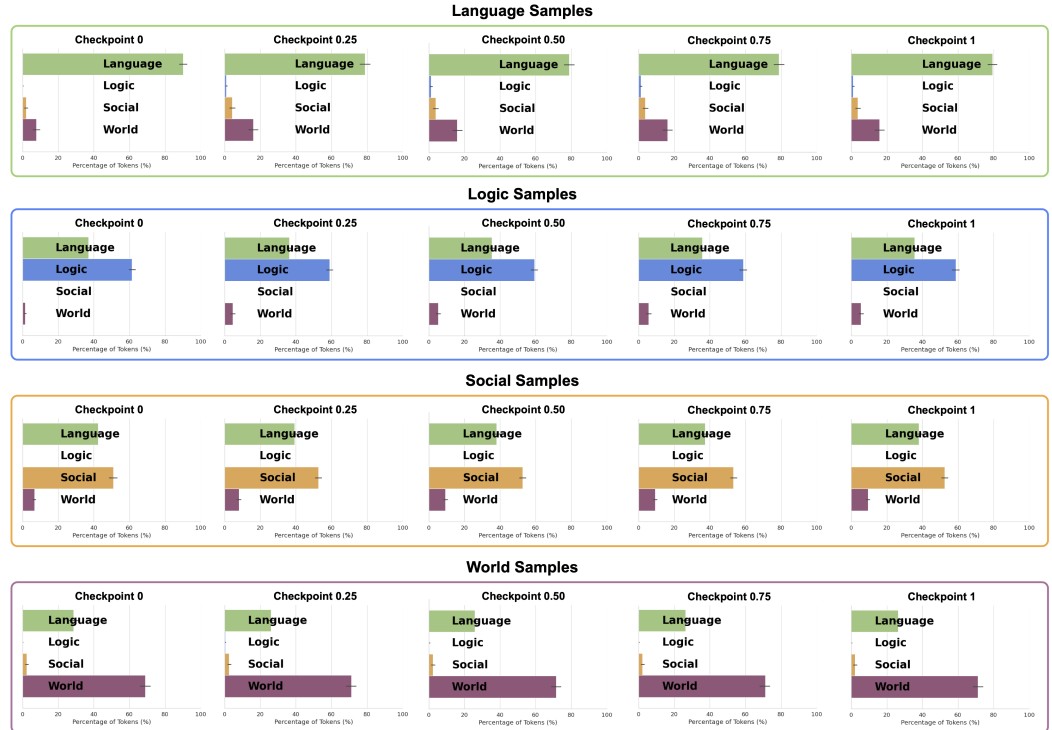

Figure 19: **Specialization Remains Consistent Throughout Training.** Token routing across checkpoints during Stage 3 training of MICRO-LLAMA-1B on the samples generated to probe each corresponding expert. Checkpoint 0 corresponds to the final weights from Stage 2. The plot shows that expert assignments remain stable throughout training, with minimal variation, indicating that the model retains its learned specialization despite the absence of explicit constraints.

Table 13: **Benchmark Results for Llama-3.1-8B Model Variants.** Performance comparison of LLAMA-3.1-8B variants across multiple benchmarks. Ablation refers to selectively removing the least relevant expert per benchmark.

| Model Name | GSM8K | BBH | MMLU | MMLU Pro | MATH | HellaSwag | PIQA | ARC-E | ARC-C | Avg |
|---|---|---|---|---|---|---|---|---|---|---|
| Llama-8B-Dense | 71.3 | 54.2 | 51.6 | 24.6 | 21.5 | 72.8 | 78.3 | 74.3 | 44.1 | 54.7 |
| Llama-8B-MoB | 69.3 | 54.1 | 52.0 | 23.5 | 21.3 | 72.2 | 78.9 | 74.6 | 46.9 | 54.8 |
| Llama-8B-MoB (Ablation) | 70.0 | 55.9 | 54.1 | 36.3 | 21.6 | 72.4 | 79.3 | 75.0 | 46.9 | 56.8 |
| MiCRo-Llama-8B | 69.1 | 53.6 | 50.7 | 23.1 | 18.1 | 72.7 | 78.7 | 75.7 | 45.9 | 54.2 |
| MiCRo-Llama-8B (Ablation) | 69.1 | 55.0 | 52.7 | 36.3 | 18.2 | 73.1 | 78.7 | 76.3 | 46.7 | 56.2 |

signed-rank test across the nine benchmarks, we find no statistically significant difference between MiCRo and either of the two baselines. The DENSE vs. MICRO comparison yields a test statistic of 31.0 (p = 0.18), and the MOB vs. MICRO comparison yields 33.5 (p = 0.10), both well above the conventional 0.05 significance threshold. However, when we ablate the most detrimental expert per task for both MiCRo and MoB we see a jump in performance, as also illustrated in Figure 7. In general, the Llama-8B experiments confirm our initial results that MICRO remains competitive on a suite of benchmarks while remaining interpretable and relevant to cognitive neuroscience.

# N    LARGE LANGUAGE MODEL USAGE

We used large language models (LLMs) solely for editing and grammatical refinement of the manuscript. All substantive ideas, analyses, and conclusions presented in this work are our own.

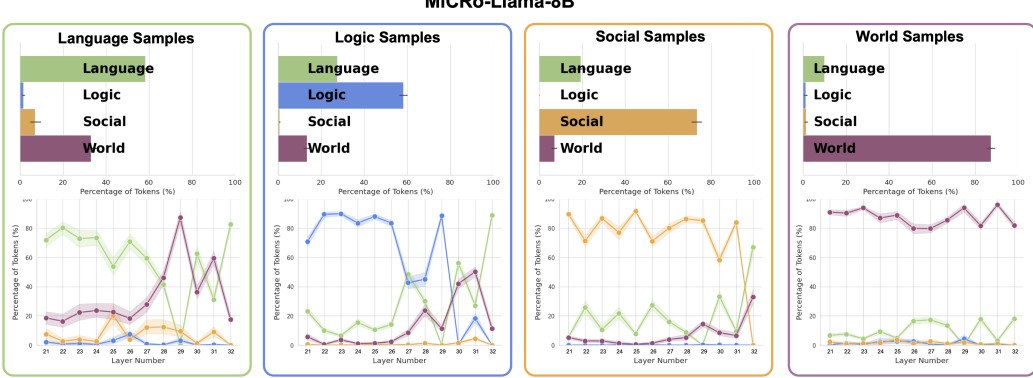

Figure 20: **Token Routing Patterns for MICRO-LLAMA-8B**. (Top) The percentage of tokens routed to each expert aggregated across the last 12 layers of MICRO-LLAMA-8B. The samples are GPT-5 generated question-answer pairs targeting specific domains. (Bottom) The corresponding layer-wise token routing patterns.

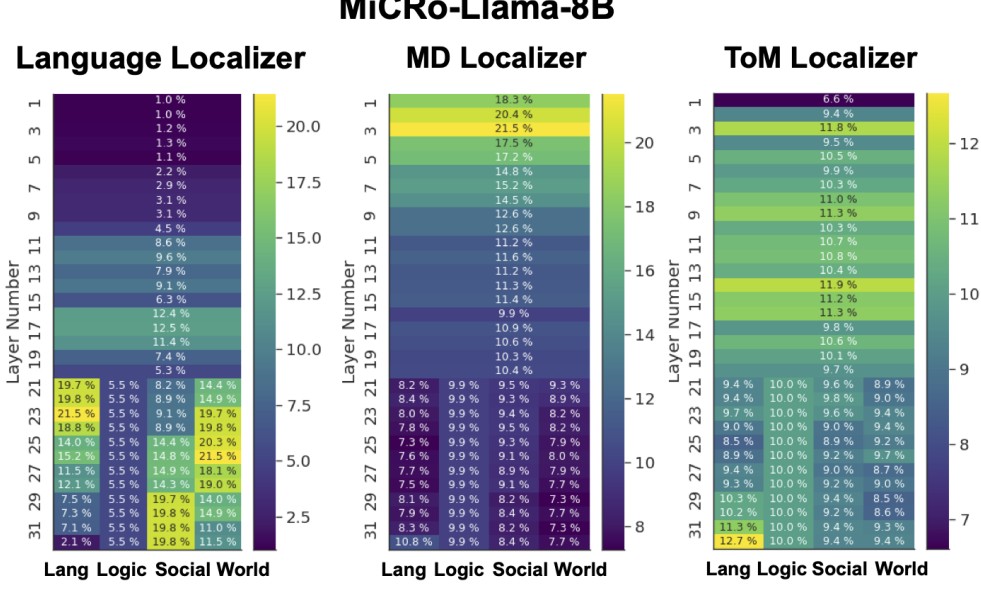

Figure 21: **Neuroscience Localizers Fail to Recover Experts For Hybrid MICRO-Llama-8B** Following the procedure in Figure 5, we apply three neuroscience-inspired functional localizers to MICRO-LLAMA-8B, which places experts only in the final 12 layers due to compute constraints. In contrast to the full MICRO variants with experts in every layer, the localizers are unable to reliably recover the expected expert specializations in this hybrid configuration.

**Prompt**

If two people disagree on how to split $10 fairly, what should they do?

**All Experts**

If two people disagree on how to split $10 fairly, they should try to find a compromise that satisfies both parties. This could involve discussing the different ways of splitting the money and finding a solution that is fair to both of them. It's important to remember that the goal is to reach a fair and equal agreement, so both parties should be willing to work together to find a solution.

**Logic**

If two people disagree on how to split $10 fairly, they should try to find a compromise that both can agree on. They can do this by trying to find a number that both people can agree on, such as the average of the two numbers, or by trying to find a number that is closer to both numbers than either number itself.

**Social**

If two people disagree on how to split $10 fairly, they should try to find a compromise that satisfies both parties. Here are some steps they can take:

1. **Communicate openly**: Both parties should express their opinions and listen to each other's perspectives. This will help them understand each other's viewpoints and find common ground.

2. **Find common ground**: Look for areas of agreement or values that both parties share. This will help them find a compromise that is fair to both of them.

3. **Discuss the reasons behind their disagreement**: Ask both parties why they disagree on how to split $10 fairly. This will help them understand each other's perspectives and find a solution that satisfies both parties.

4. **Explore alternative solutions**: If they cannot find a compromise that satisfies both parties, they can explore alternative solutions, such as splitting the money in a different way or finding a different way to split the money fairly.

5. **Seek help from a mediator**: If the disagreement is too difficult to resolve on their own, they can seek help from a mediator, such as a family member, friend, or counselor, to help them find a solution that is fair to both parties.

6. **Find a solution that works for both parties**: Ultimately, they should find a solution that works for both parties and is fair to both of them. This may involve compromising on certain points or finding a different way to split the money fairly.

**World**

If two people disagree on how to split $10 fairly, they should consider the following options:

1. Mediation: One person can mediate the disagreement by suggesting a compromise or a fair split. This can help both parties reach an agreement without further conflict.

2. Negotiation: Both parties can negotiate a fair split by discussing the benefits and drawbacks of different splits. They can try to find a solution that both parties can agree on.

3. Arbitration: If the disagreement is significant and cannot be resolved through mediation or negotiation, both parties can seek the help of a neutral third party, such as a mediator or arbitrator, to help them reach an agreement.

4. Legal action: If the disagreement is not resolved through mediation or negotiation, both parties can consider seeking legal advice or taking legal action to enforce their rights.

5. Consultation: Both parties can consult with a neutral third party, such as a financial advisor or mediator, to help them reach a fair split. This can provide a neutral perspective and help both parties reach an agreement.

Ultimately, the best course of action depends on the specific circumstances of the disagreement and the preferences of the parties involved.

Figure 22: **Example for Steering Model Behavior by Expert Ablation.** Responses of MICRO-LLAMA-3B to the given prompt when only the target expert and the language expert are retained. The differences illustrate the causal role of each expert and demonstrate how ablations can steer the model's behavior.

**Prompt**

If your friend asks whether you like their new haircut, but you don't, what should you say?

**All Experts**

You can say something like, "I appreciate your creativity, but I don't think this is the style for me."

**Logic**

You can say something like, "I'm glad you're happy with it, but I'm not a fan of it myself."

**Social**

When your friend asks if you like their new haircut, but you don't, it's important to be honest and respectful. Here's a suggestion on how to respond:

1. **Be Honest**: You can say something like, "I really appreciate your style, but this particular haircut isn't for me. I think it's a great choice for you, though."

2. **Compliment**: If you do like something about their new look, make sure to compliment it. For example, "I love your hair color. This new cut just isn't for me."

3. **Express Your Feelings**: If you're not a fan of the haircut, you can say something like, "I just don't think this particular style works for me. I'm sure it's going to be a hit with you."

4. **Offer Support**: If your friend is disappointed, offer your support. "I know you're disappointed, but I'm sure you'll find something you love."

Remember, it's important to be kind and understanding, even if you don't like the haircut. It's also okay to be honest about your feelings.

**World**

You can say something like, "I'm sorry, but I don't really like it."

Figure 23: **Example for Steering Model Behavior by Expert Ablation.** Responses of MICRO-LLAMA-3B to the given prompt when only the target expert and the language expert are retained. The differences illustrate the causal role of each expert and demonstrate how ablations can steer the model's behavior.

**Prompt**

Your friend is late to a meeting. Why might that be?

**All Experts**

Your friend might be late to a meeting for a variety of reasons. They could be running late, they might have forgotten their keys, or they could be in a rush to get to the meeting. It's important to remember that it's not always possible to avoid being late, and it's important to respect your friend's time and space.

**Logic**

Your friend might be late to a meeting because they are late to work, or they are late to the bus, or they are late to the train, or they are late to the airport...

**Social**

Your friend might be late to a meeting for various reasons. They could be late due to a personal emergency, they might be late because they are not ready to meet, or they could be late due to a lack of preparation. It's important to approach this situation with empathy and understanding.

**World**

Your friend might be late to a meeting for a variety of reasons. They could be running late, they might be waiting for someone else, or they could be experiencing a delay in their usual schedule. It's important to remember that being late can be frustrating and may not be a good reflection of your friend's character or reliability.

Figure 24: **Example for Steering Model Behavior by Expert Ablation.** Responses of MICRO-LLAMA-1B to the given prompt when only the target expert and the language expert are retained. The differences illustrate the causal role of each expert and demonstrate how ablations can steer the model's behavior.

**Prompt**

How could two people decide who gets the last slice of pizza?

**All Experts**

Two people could decide who gets the last slice of pizza by using a fair and unbiased method. Here are a few suggestions:

1. Random draw: One person could draw a random number between 1 and 2, and the other person would get the last slice if their number is 1.

2. Coin flip: One person could flip a coin, and the other person would get the last slice if the coin lands on heads.

3. Rolling dice: One person could roll a pair of dice, and the other person would get the last slice if the sum of the dice is even.

4. Drawing cards: One person could draw a card from a deck, and the other person would get the last slice if the card is a number between 1 and 2.

5. Using a fair method: One person could use a fair method, such as a coin flip or a random draw, to determine who gets the last slice of pizza.

It is important to ensure that the method used to determine who gets the last slice of pizza is fair and unbiased, so that both people feel that the decision is made in good faith.

**Logic**

To decide who gets the last slice of pizza, you can use a variety of methods. One way is to take turns picking the last slice. Another way is to flip a coin or use a random method to determine who gets the last slice. Remember, it's important to respect each other's choices and enjoy the pizza together.

**Social**

When deciding who gets the last slice of pizza, it's important to consider the following factors:

1. **Relationship and history**: If you have a history of sharing pizza or have a close relationship with the other person, they may be more likely to share the last slice with you.

2. **Current mood and energy**: If you're feeling particularly energetic or in a good mood, you may be more likely to share the last slice of pizza with the other person.

3. **Physical appearance**: If you're physically more capable or have a better sense of balance, you may be more likely to share the last slice of pizza with the other person.

4. **Emotional state**: If you're feeling particularly happy or content, you may be more likely to share the last slice of pizza with the other person.

5. **Situational context**: If you're in a particular situation or context, such as being in a group or having a specific reason for sharing pizza, you may be more likely to share the last slice of pizza with the other person.

6. **Nonverbal cues**: If you're using nonverbal cues, such as a smile or a friendly gesture, you may be more likely to share the last slice of pizza with the other person.

7. **Communication**: If you're communicating with the other person, you may be able to convey your intentions or desires more effectively, which could lead to a more positive outcome.

8. **Physical proximity**: If you're physically closer to the other person, you may be more likely to share the last slice of pizza with them.

9. **Personal values**: If you have a strong sense of fairness or equality, you may be more likely to share the last slice of pizza with the other person.

10. **Social norms**: If you're in a particular social group or community, there may be established norms or expectations regarding sharing

**World**

Your friend might be late to a meeting for a variety of reasons. They could be running late, they might be waiting for someone else, or they could be experiencing a delay in their usual schedule. It's important to remember that being late can be frustrating and may not be a good reflection of your friend's character or reliability.

Figure 25: **Example for Steering Model Behavior by Expert Ablation.** Responses of MICRO-LLAMA-3B to the given prompt when only the target expert and the language expert are retained. The differences illustrate the causal role of each expert and demonstrate how ablations can steer the model's behavior.

