# OpenReview forum: "Mixture of Cognitive Reasoners: Modular Reasoning with Brain-Like Specialization"
_ICLR.cc/2026/Conference — ICLR 2026 Poster_

### Official Review · Reviewer_D255 · 2025-10-18

**Soundness:** 3
**Presentation:** 2
**Contribution:** 3
**Rating:** 4
**Confidence:** 3

**Summary:**

This paper proposes Mixture of Cognitive Reasoners (MICRO), a modular transformer architecture inspired by the brain’s functional specialization into distinct cognitive networks (language, logic, social reasoning, world knowledge). The model partitions each transformer block into four “experts” corresponding to these domains, trained through a three-stage curriculum intended to induce brain-like specialization. The authors claim this architecture enhances interpretability, controllability, and alignment with human behavioral benchmarks (COGBENCH), while maintaining competitive reasoning performance on standard NLP tasks (e.g., GSM8K, MATH, MMLU, BBH).

**Strengths:**

The motivating analogy to cognitive neuroscience is clear and interesting: the authors connect transformer modularity to the brain’s distributed but specialized functional architecture.

The paper is technically ambitious and proposes a relatively clean experimental pipeline (three-stage training) that is easy to reproduce conceptually.

The authors provide a comprehensive empirical evaluation, including behavioral alignment metrics, neuroscience “localizers,” and ablation analyses.

The inclusion of interpretable routing and causal ablation is a meaningful step toward testable hypotheses about functional decomposition in large models.

**Weaknesses:**

1. Conceptual clarity and motivation.
While the analogy to brain modularity is compelling, the paper does not clearly articulate why such modular specialization is desirable in language models. The claimed benefits—interpretability and controllability—are asserted but not demonstrated. The architecture yields mixed performance gains, suggesting that interpretability alone may not justify the added complexity.

2. Neuroscientific grounding.
The mapping between the four “expert” modules and the purported brain networks is overly categorical and simplified. The cognitive neuroscience literature remains divided on several of these assumptions. For example:
- The supposed separation between language and reasoning networks remains debated;
- The distinction between “logic” and “math” is underdefined, yet the paper treats them as part of a single module.
- Recent work (e.g., Hope Kean et al.) suggests that logical reasoning may rely on a distinct neural network separate from the multiple-demand system invoked here.
As such, the neuroscience framing may be more metaphorical than mechanistic.

3. Data labeling and methodology.
The “MICRO_SFT” dataset is central to inducing specialization, but the criteria for domain labeling are unclear. The dataset was pseudo-labeled using O1 and GPT-4o, but there is little evidence that these models’ judgments correspond to meaningful domain boundaries. No human validation or inter-rater reliability is reported. It remains uncertain whether the apparent “specialization” reflects genuine cognitive decomposition or artifacts of the labeling pipeline.

4. Relevance to cognitive neuroscience.
The authors seem to suggest (implicitly) that MICRO could bridge AI and brain science, but the connection is speculative. The architecture may be inspired by brain modularity, yet it does not provide new neuroscientific insight—no neural data are modeled, and the alignment tests (using functional localizers) are correlational. While ablation studies show that removing experts affects performance, this is a coarse-grained effect and not obviously interpretable at the cognitive level.

**Questions:**

How do you justify the specific choice of four networks, given the ongoing debates about their boundaries and overlap in the brain?

What validation steps were taken to ensure the O1/GPT-4o pseudo-labels correspond to human-like task domains?

How do you distinguish interpretability (as in mechanistic insight) from mere architectural labeling?

What concrete cognitive-neuroscience hypotheses does MICRO make that could be tested empirically?

Given the small size of the MICROSFT dataset (≈3k examples), how sensitive are your results to its composition or labeling noise?

---

> ### Author Response · Authors · 2025-11-21
>
> We thank the reviewer for their feedback, and for finding the paper technically ambitious but with a relatively clean experimental pipeline that is easy to reproduce conceptually. We appreciate the reviewer’s acknowledgement of our comprehensive empirical evaluation and for the impact of our work toward testable hypotheses about functional decomposition in large models.
> We address the reviewer’s comments below.
>
> **Re Weaknesses**
>
> **1/**
> > why such modular specialization is desirable in language models
>
> Achieving an interpretable, modular architecture without sacrificing performance is non-trivial. In many settings, interpretability and performance are viewed as a trade-off, where introducing structure or constraints typically reduces capability. Our results demonstrate that this trade-off is not inevitable. MiCRo produces stable, interpretable, and causally meaningful specialization (as shown by the routing patterns in Figures 3, 12, 13, 14, 19 and the expert ablations in Figures 4, 16) while maintaining competitive performance on reasoning benchmarks and strong alignment with human behavior.
>
> This directly illustrates why modular specialization is desirable:
>
> - It yields transparent computation, where different parts of the model systematically handle different cognitive domains;
> - It enables controllability, as shown in Figure 4 and the Figures associated with Appendix K where selectively activating or ablating experts steers the model’s reasoning in a mechanistic way; and
> - It provides a meaningful bridge to cognitive neuroscience, allowing us to recover experts using established functional localizers, an approach previously linked to neural predictivity (AlKhamissi et al., 2025).
>
> Importantly, modularity offers practical performance advantages as well. MiCRo (Ablation) achieves improvements over dense baselines by removing experts that are detrimental for a given benchmark. For instance, selectively ablating the social expert on mathematical reasoning tasks yields significant performance gains (Figure 4). This kind of targeted mechanistic intervention demonstrates an additional, concrete benefit of modular specialization.
>
> That said, while MiCRo does not uniformly outperform dense models, MiCRo introduces interpretability and controllability without paying a performance penalty, while offering new avenues for mechanistic insight, behavioral alignment, and neuroscientific testing.
>
> -----
>
> **2/**
> > The mapping between the four “expert” modules and the purported brain networks is overly categorical and simplified.
>
> We agree that the mapping between linguistic, logical/MD, social/ToM, and world/DMN networks in the human brain is an active area of debate. Our goal is not to assert a definitive taxonomy of cognitive networks, but to test whether ideas proposed in the cognitive neuroscience literature can provide a useful inductive bias for modular learning in LLMs, yielding interpretable routing patterns while maintaining strong performance on reasoning benchmarks.
>
> We intentionally chose a coarse but well-studied partition, grounded in decades of results showing:
>
> - a selective language network (Fedorenko et al. 2010; 2011),
> - a broadly domain-general multiple-demand (MD) network supporting reasoning (Duncan & Owen, 2000; Duncan 2010; Fedorenko et al., 2013).
> - a selectively engaged social/ToM network (Gallagher et al., 2000; Saxe & Kanwisher, 2003; Saxe & Powell, 2006).
> - a default mode network associated with world knowledge, narrative structure, and situation modeling (Gusnard et al., 2001; Buckner et al., 2008; Buckner & DiNicola, 2019).
>
> The goal was not to capture all cognitive systems exhaustively, but to test whether coarse modularity, inspired by these well-replicated distinctions, leads to stable and interpretable specialization in transformers.
>
> We also fully agree that the boundaries of these networks are graded, not categorical, and brain regions interact with one another during any cognitive task. Importantly, MiCRo also exhibits this behavior: tokens may be routed to different experts across layers, and tokens processed by different experts attend to one another.
>
> ----
>
> Our response continues in the next comment.

---

> > ### Author Response · Authors · 2025-11-21
> >
> > **Weaknesses (cont.)**
> >
> > **3/** & **Re Question #5**
> > > No human validation or inter-rater reliability is reported on the MiCRo_SFT dataset. It remains uncertain whether the apparent “specialization” reflects genuine cognitive decomposition or artifacts of the labeling pipeline.
> >
> > The reviewer raises an important point about labeling quality. We respond on several fronts:
> >
> > (i) **Router training is unsupervised.**
> >
> > The router is not trained using domain labels; instead, it is optimized solely through a next-token prediction loss. In Stage-2, it learns to select the appropriate expert based on the inductive bias established in Stage-1. Therefore, the observed specialization does not stem from memorizing labels or from any noise in the pseudo-labeling process.
> >
> > (ii) **We evaluate specialization on independent held-out datasets.**
> >
> > In Figures 12-14 in the Appendix, we show MiCRo’s routing on independently collected benchmarks, including Minerva Math, MMLU, BBH, and synthetic domain-controlled datasets, aligns with intuitive domain boundaries. This strongly suggests that specialization is not an artifact of the MiCRo_SFT pseudo-labels.
> >
> > (iii) **Human validation.**
> >
> > Following the reviewer’s suggestion, we asked three human annotators to label a subset of MiCRo_SFT (n=253). We include the full results and inter-rater agreement statistics in Appendix B in the revised PDF (Tables 2-6). We also show below a summary of the agreement statistics. Overall, the dataset shows moderate inter-annotator consistency across all metrics, with variation typical of multi-class subjective labeling tasks. The LLM’s labels align with human judgments at levels comparable to human–human agreement, and the model agrees most strongly with annotators H1 and H2, who themselves exhibit the highest mutual consistency.
> >
> > ---
> >
> > ## **Human–Human Agreement (3 Annotators)**
> >
> > * **Krippendorff’s alpha (nominal, 3 humans):** 0.517
> > * **Fleiss’ kappa (3 humans):** 0.517
> > * **3-way percent agreement (all annotators agree):** 49.4%
> >
> > **Pairwise Agreement:**
> >
> > * **H1–H2:**
> >
> >   * Percent agreement: 76.3%
> >   * Cohen’s kappa: 0.681
> > * **H1–H3:**
> >
> >   * Percent agreement: 58.9%
> >   * Cohen’s kappa: 0.431
> > * **H2–H3:**
> >
> >   * Percent agreement: 58.5%
> >   * Cohen’s kappa: 0.446
> >
> > ---
> >
> > ## **Human–LLM Agreement**
> >
> > * **Items with unique human majority:** 240
> > * **Items with ties:** 13
> >
> > ### **LLM vs. Human Majority Vote**
> >
> > * **Accuracy:** 0.658
> > * **Macro F1:** 0.666
> > * **Cohen’s kappa:** 0.533
> >
> > ### **LLM as a 4th Annotator**
> >
> > * **Krippendorff’s alpha (nominal, 3 humans + LLM):** 0.497
> >
> > **Pairwise Agreement with LLM:**
> >
> > * **H1–LLM:**
> >
> >   * Percent agreement: 62.8%
> >   * Cohen’s kappa: 0.489
> > * **H2–LLM:**
> >
> >   * Percent agreement: 62.5%
> >   * Cohen’s kappa: 0.492
> > * **H3–LLM:**
> >
> >   * Percent agreement: 59.7%
> >   * Cohen’s kappa: 0.448
> >
> > ---
> >
> > **4/** & **Re Question #4**
> > > Relevance to cognitive neuroscience.
> >
> > MiCRo makes concrete and testable predictions about how different cognitive systems work together to support a given behavior. Here we focus on one example: how information is routed across time and across brain areas—an area of neuroscience that is challenging to probe with current neuroimaging methods, making a priori model‑based hypotheses particularly valuable.
> >
> > **Temporal routing of information.**
> >
> > **Hypothesis.** When humans solve multi‑step reasoning tasks such as GSM8K‑style math word problems, cortical networks are engaged in the same sequence that MiCRo’s experts are recruited (Fig. 3f):
> > Early language network engagement during problem reading → Increased world/DMN engagement as a situation model is built → Stronger logic/MD engagement during calculation → Answer formulation
> >
> > MiCRo therefore predicts specific trajectories of information transfer across time and across known brain networks.
> >
> > **Empirical tests.** Temporal routing of information can be tested via time‑resolved neuroimaging—(M)EEG/ECoG while participants solve math word problems. For each token or time window, MiCRo’s layer‑wise routing weights over experts can be extracted and we can test whether the predicted routing sequence best explains the time‑course of activation in independently localized language, DMN, and MD networks. Further, we can make concrete predictions and tests about how this routing sequence shifts when the math world problems are manipulated (e.g., increased world‑knowledge demands vs. pure arithmetic demands).
> >
> > Thus, while the architecture is inspired by the brain, it also produces concrete hypotheses that can guide new experiments.
> >
> > ----
> >
> > Our response continues in the next comment.

---

> > > ### Author Response · Authors · 2025-11-21
> > >
> > > **Re Questions**
> > >
> > > **1/**
> > > > How do you justify the specific choice of four networks, given the ongoing debates about their boundaries and overlap in the brain?
> > >
> > > We selected these four domains because they correspond to the widely replicated functional networks that are diverse enough to be relevant to standard NLP benchmarks and because they offer a clear and interpretable initial partition.
> > >
> > > MiCRo does not claim these are the only networks, nor the optimal partition. Rather, they provide a tractable starting point for evaluating brain-inspired modularity.
> > >
> > > Future work can incorporate additional experts (e.g., intuitive physics, logical reasoning) as suggested by recent studies.
> > >
> > > ----
> > >
> > > **2/**
> > > > What validation steps were taken to ensure the O1/GPT-4o pseudo-labels correspond to human-like task domains?
> > >
> > > As noted above:
> > > - The router is not trained on labels, reducing sensitivity to label errors.
> > > - Routing patterns generalize to independent datasets.
> > > - Human validation confirms pseudo-labels align well with human judgments.
> > > - Specialization persists through Stage-3, indicating robustness.
> > >
> > > ----
> > >
> > > **3/**
> > > > How do you distinguish interpretability (as in mechanistic insight) from mere architectural labeling?
> > >
> > > MiCRo’s interpretability does not come from assigning architectural labels to experts, but from empirical, mechanistic evidence that these experts acquire distinct functional roles. First, routing patterns emerge naturally—our router is not trained on domain labels—yet tokens consistently route to the appropriate expert on held-out datasets (Figures 3, 12-14). Second, causal ablations show domain-specific impairment (e.g., logic ablation harms GSM8K; Figure 4, 16), confirming that each expert supports a distinct computation. Third, functional localizers from neuroscience independently recover the same experts (Figure 5), indicating that specialization is not an artifact of the architecture. Finally, multi-domain questions recruit multiple experts across layers, demonstrating functional depth rather than static labeling.
> > >
> > > Together, these results show that MiCRo’s specialization is mechanistically interpretable, not merely architecturally labeled.
> > >
> > > ----
> > >
> > > **4/**
> > > > What concrete cognitive-neuroscience hypotheses does MICRO make that could be tested empirically?
> > >
> > > Please see our response to Weakness #4.
> > >
> > > ----
> > >
> > > **5/**
> > > > Given the small size of the MICRO_SFT dataset (≈3k examples), how sensitive are your results to its composition or labeling noise?
> > >
> > > Please see our response to Weakness #3.
> > >
> > > -----
> > >
> > > **Conclusion**
> > >
> > > We thank the reviewer again for the insightful feedback. We have incorporated additional clarifications, human validation, and expanded neuroscientific framing in the revised manuscript, which we believe significantly strengthen the contribution of the paper.

---

> > > > ### Comment · Reviewer_D255 · 2025-11-25
> > > >
> > > > Dear authors,
> > > >
> > > > Thank you for your extensive response. I enjoyed the paper; it was one of the best I reviewed. I will update the score then.

---

> > > > > ### Author Response · Authors · 2025-11-26
> > > > >
> > > > > Dear Reviewer,
> > > > >
> > > > > Thanks again for your thoughtful feedback, we really appreciate your positive comments about the paper. We noticed the score hasn’t been updated yet, so we just wanted to check in in case it was missed. If there’s anything else you’d like us to clarify or address, we’d be more than happy to.

---

### Official Review · Reviewer_A5jV · 2025-10-28

**Soundness:** 2
**Presentation:** 3
**Contribution:** 3
**Rating:** 4
**Confidence:** 3

**Summary:**

This paper proposes a method (MiCRo) for post-training a transformer to incorporate specialized expert blocks across four domains: language, logic, social, and world knowledge. Inspired by the functional regions observed in brain networks, the approach involves a three-stage training process: (1) duplicating the original transformer blocks to create four expert blocks, each trained on data labeled at the sentence level according to domain; (2) freezing the network parameters and training routers to direct inputs to the appropriate expert blocks; and (3) conducting large-scale supervised fine-tuning. Experimental results demonstrate that, after training, these expert blocks exhibit distinct and specialized functionalities.

**Strengths:**

1.	The proposed method is well-motivated by the four modular functional regions in human brain.

2.	The paper is clearly written and easy to follow.

**Weaknesses:**

1.	The diversity and scale of tested models can be further enriched. The paper test Llama, SmolLM, and OLMO models of up to 3B parameters. I would recommend testing on 7B scale models from other model families, such as Qwen2.5-7B-Instruct. Small models sometimes behave very different from large models, so experiments on 7B scale models would make the results more convincing.

2.	The paper categorizes MMLU as a reasoning benchmark, which is not quite accurate. I would recommend extra experiments on MMLU-Pro (or a subset of it), which contains more reasoning-oriented questions.

3.	In Figure 7 and Table 3, the proposed MiCRo architecture does not outperform the dense model baseline on reasoning tasks, even though it has three times more parameters. While MiCRo-Ablation (which shows the best results when up to one expert is ablated) can sometimes achieve better performance than the dense model baseline, it requires four times more compute at test time. Therefore, the improvement of MiCRo on reasoning tasks is still limited.

**Questions:**

1.	The paper notes that small MOE models do not show expert specialization like the MiCRo counterpart. Is it due to the limitation of the model scale?

2.	If a question is related to more than one field (e.g., a question that is both related to math and social science), what will be the behavior of the MiCRo model? Are there any experiments on this?

3.	The paper mentions that using a small amount of data in Stage 1 and 2 suffices to elicit expert specialization behavior and that this specialization remains in the large-scale SFT stage. What is the intuition behind this phenomenon?

4.	In some subplots in Figure 4 (e.g., MiCRo-Llama-3B on MMLU_other), ablating any of the logic, social, or world expert can improve performance. It seems weird since it implies all of these experts are detrimental to the task. Are there any explanations on this phenomenon?

5.	The meaning of the marks “*” and “ns” in Figure 7 is not clear. One can guess that it shows the significance of the Welch’s t-tests, but it is better to explain them in the caption.

---

> ### Author Response · Authors · 2025-11-21
>
> We thank the reviewer for their constructive feedback and for recognizing that the proposed method is well-motivated by the four modular functional regions in the human brain, as well as for noting that the paper is clearly written and easy to follow.
> We address each of the reviewer’s comments below.
>
> **Re Weaknesses**
>
> **1/**
> > scaling up experiments on 7B scale models would make the results more convincing.
>
> Following the reviewer’s suggestion, we have post-trained three Llama-3.1-8B variants:
> (1) MiCRo-Llama-8B, (2) its modular baseline Llama-8B-MoB, and (3) Llama-8B-Dense.
> Due to compute and time constraints, we instantiated experts only in the last 12 layers of MiCRo-Llama-8B and Llama-8B-MoB, keeping the earlier layers dense. This choice is motivated by our prior findings (Figures 13–14, Appendix) showing that early layers predominantly route to language experts, while non-language specializations emerge in later layers. The results confirm that MiCRo-Llama-8B exhibits the expected routing patterns in its last 12 layers (Figure 20 in the revised PDF), and we include the results on the reasoning benchmarks in the table below (including MMLU-Pro as per Reviewer A5jV suggestion) and in the revised manuscript (Appendix M; Table 13).
>
> The table below shows the results on the 9 benchmarks of the three Llama-3.1-8B model variants, along with the results when we remove the most detrimental expert across all layers for a given task for the MiCRo and MoB model variants, which we denote by appending (Ablation), similar to Figure 7 in the main paper. Using a paired Wilcoxon signed-rank test across the nine benchmarks, we find no statistically significant difference between MiCRo and either of the two baselines. The Dense vs. MiCRo comparison yields a test statistic of 31.0 (p = 0.18), and the MoB vs. MiCRo comparison yields 33.5 (p = 0.10), both well above the conventional 0.05 significance threshold. However, when we ablate the most detrimental expert per task for both MiCRo and MoB we see a jump in performance, as also illustrated in Figure 7.
>
> In general, the Llama-8B experiments confirm our initial results that MiCRo remains competitive on a suite of benchmarks while remaining interpretable and relevant to cognitive neuroscience as discussed in our response to Reviewer D255 (Weakness #4).
>
> | Model Name                | GSM8K | BBH | MMLU | MMLU Pro | MATH  | HellaSwag | PIQA | ARC Easy | ARC Challenge | Average |
> |---------------------------|-------|--------------|----------------|--------------------|----------------|------------|------|----------|----------------|---------|
> | Llama-8B-Dense            | 71.3  | 54.2         | 51.6           | 24.6               | 21.5           | 72.8       | 78.3 | 74.3     | 44.1           | 54.7    |
> | Llama-8B-MoB              | 69.3  | 54.1         | 52.0           | 23.5               | 21.3           | 72.2       | 78.9 | 74.6     | 46.9           | 54.8    |
> | Llama-8B-MoB (Ablation)   | 70.0  | 55.9         | 54.1           | 36.3               | 21.6           | 72.4       | 79.3 | 75.0     | 46.9           | 56.8    |
> | MiCRo-Llama-8B            | 69.1  | 53.6         | 50.7           | 23.1               | 18.1           | 72.7       | 78.7 | 75.7     | 45.9           | 54.2    |
> | MiCRo-Llama-8B (Ablation) | 69.1  | 55.0         | 52.7           | 36.3               | 18.2           | 73.1       | 78.7 | 76.3     | 46.7           | 56.2    |
>
> -----------------------------------
>
> **2/**
> > extra experiments on MMLU-Pro
>
> We appreciate the suggestion and have evaluated all of our models on MMLU-Pro, which contains more reasoning-oriented questions. We report these new results below and in the revised manuscript in Figure 7 (Main) and Table 8 (Appendix).
>
> | Base Model      | MiCRo (Ablation) | MiCRo | MoB (Ablation) | MoB  | Dense |
> |-----------------|------------------|-------|------------------|------|-------|
> | Llama-3.2-1B    | 14.4             | 10.7  | 14.2             | 11.0 | 11.2  |
> | Llama-3.2-3B    | 32.8             | 19.0  | 34.0             | 19.1 | 19.6  |
> | Llama-3.1-8B    | 36.3             | 23.1  | 36.3             | 23.5 | 24.6  |
> | SmoLLM2-135M    | 11.3             | 7.9   | 11.0             | 7.4  | 7.8   |
> | SmoLLM2-360M    | 11.0             | 10.1  | 12.0             | 9.8  | 9.9   |
> | OLMo-2-1B       | 22.4             | 14.9  | 20.8             | 14.9 | 14.0  |
> | **Average**     | **21.4**         | 14.3  | **21.4**         | 14.3 | 14.5  |
>
> Overall, we find that MiCRo performs on par with the MoB and Dense baselines across all base models, while being interpretable, and the ablation configuration yielding substantial performance gains, which we discuss more in the next comment.
>
> -----------------------------
>
> Our response continues in the next comment.

---

> > ### Author Response · Authors · 2025-11-21
> >
> > **Weaknesses (cont.)**
> >
> > **3/**
> > > MiCRo-Ablation can sometimes achieve better performance than the dense model baseline, it requires four times more compute at test time. Therefore, the improvement of MiCRo on reasoning tasks is still limited.
> >
> > We would like to clarify that MiCRo-Ablation does not necessarily require 4× more compute at inference. Because MiCRo’s experts are interpretable and causally meaningful, we can identify the irrelevant expert based on our understanding of the benchmark’s domain. We did that for GSM8K and Minerva Math. To automate this, a simple 2-pass procedure suffices:
> >
> > - The first pass accumulates router probabilities to determine the least-used expert on a specific task (e.g., social expert for mathematical reasoning tasks).
> > - The second pass ablates that expert during evaluation (e.g., remove all social experts).
> >
> > We verified (on a subset of the benchmarks) that this procedure yields the same results as expected.
> >
> > In contrast, for the MoB baseline, we performed five separate runs (ablating each expert once plus the intact model) and reported the maximum performance across these runs. This was necessary because the MoB model’s routing patterns show no consistent specialization trends. To make MiCRo comparable with MoB for benchmarks that do not have one dominant domain (such as MMLU and BBH), we performed a similar analysis. However, this could be simplified using the 2-pass procedure mentioned above.
> >
> > --------------------
> >
> >
> > **Re Questions**
> >
> > **1/**
> > > The paper notes that small MOE models do not show expert specialization like the MiCRo counterpart. Is it due to the limitation of the model scale?
> >
> > This limitation likely stems from insufficient representational capacity in smaller MoE models. If a small model cannot solve tasks from a given domain (e.g., theory-of-mind questions), the router lacks a signal to distinguish between domains, leading to uniform routing. MiCRo, by contrast, includes domain-specific multi-head attention within each expert, allowing even smaller models to differentiate more effectively across domains.
> >
> > -----
> >
> > **2/**
> > > If a question is related to more than one field, what will be the behavior of the MiCRo model?
> >
> > As illustrated in Figure 1(d), routing in MiCRo occurs at the token level. Each token is routed to its most relevant expert, meaning that multi-domain questions engage different experts for different parts of the input: e.g., tokens requiring mathematical reasoning are processed by the logic expert, while socially contextual tokens are routed to the social expert.
> > Importantly, each token can be routed to different experts across layers within a single forward pass, and tokens processed by different experts still attend to one another through the attention mechanism. This enables continuous cross-expert communication.
> >
> > -----
> >
> > **3/**
> > > small amount of data in Stage 1 and 2 suffices to elicit expert specialization behavior and that this specialization remains in the large-scale SFT stage. What is the intuition behind this phenomenon?
> >
> > The specialization established during Stages 1–2 forms a stable inductive bias that anchors the model in a meaningful local minimum. During large-scale SFT (Stage 3), this bias is maintained rather than overwritten. We verify this by tracking routing patterns across checkpoints (Figure 19, Appendix of revised version), which remain largely stable throughout Stage 3 training.
> >
> > -----
> >
> > **4/**
> > >  ablating any of the logic, social, or world expert can improve performance (e.g., MiCRo-Llama-3B on MMLU_Other). It seems weird since it implies all of these experts are detrimental to the task. Are there any explanations on this phenomenon?
> >
> > In such cases, the language expert dominates task performance, and removing other experts reduces interference. The full set of ablation results is shown in Figure 16 (Appendix), confirming the importance of language experts.
> >
> > ----
> >
> > **5/**
> > >  The meaning of the marks “*” and “ns” in Figure 7 is not clear. One can guess that it shows the significance of the Welch’s t-tests, but it is better to explain them in the caption.
> >
> > We thank the reviewer for pointing this out. As noted in the caption of Figure 7 (Line 453), we already state that “significance is assessed with pairwise Welch’s t-tests.” Nevertheless, we agree that the meaning of the “ * ” and “n.s.” markers should be made explicit. In the revised version, we clarified that these markers indicate statistical significance based on Welch’s t-tests (“ * ” = significant, “n.s.” = not significant).
> >
> > ----
> >
> > **Summary**
> >
> > We appreciate the reviewer’s feedback and have incorporated additional experiments (Llama-8B, MMLU-Pro) and clarifications in the revision. We hope these updates address the reviewer’s concerns and further strengthen the paper’s contributions.

---

> > > ### Comment · Reviewer_A5jV · 2025-11-25
> > >
> > > Thank you for the detailed response. I have an additional question regarding the response to weakness 3.
> > >
> > > The new method still requires 2x compute of the dense model baseline, so the comparison here is not controlled in terms of test-time compute. Could you try another baseline that lets the dense model rollout for 3 times and then employs majority voting for the answer? Although this baseline uses 3x test-time compute, not 2x (as used in Micro-Abaltion), it can demonstrate how increasing test-time compute can enhance the performance of dense models.

---

> > > > ### Author Response · Authors · 2025-11-26
> > > >
> > > > Thank you for the suggestion. We ran the additional comparison using self-consistency (majority vote over three rollouts) for both `Llama-1B-Dense` and `Llama-3B-Dense` on all generative benchmarks. While this improves the dense baselines, MiCRo (Ablation) still achieves higher average performance across benchmarks, indicating that the gains are not solely due to increased test-time compute. The results are shown in the table below, and we hope this addresses the remaining concern.
> > > >
> > > > | Base Model      | Model Name        | Configuration         | GSM8K | BBH  | MMLU | MATH | MMLU-Pro | Average |
> > > > |-----------------|-------------------|------------------------|-------|--------------|----------------|----------------|-------------------|---------|
> > > > | **Llama-3.2-1B** | Llama-1B-Dense     | Majority Vote (k=3)    | 33.8% | 30.3%        | 31.0%          | 5.3%           | 11.1%             | 22.3%   |
> > > > |                 | MiCRo-Llama-1B | Ablation               | 39.0% | 30.6% | 31.5% | 6.1% | 14.4% | 24.3% |
> > > > | **Llama-3.2-3B** | Llama-3B-Dense     | Majority Vote (k=3)    | 59.1% | 44.6%        | 49.0%          | 15.2%          | 20.4%             | 37.7%   |
> > > > |                 | MiCRo-Llama-3B | Ablation               | 58.0% | 42.2%        | 45.7%          | 12.7%          | 32.8%         | 38.3%   |

---

### Official Review · Reviewer_vTG9 · 2025-10-29

**Soundness:** 4
**Presentation:** 4
**Contribution:** 4
**Rating:** 6
**Confidence:** 3

**Summary:**

This paper proposes MICRO (Mixture of Cognitive Reasoners), a novel mixture-of-blocks architecture designed to induce functional specialization across different experts. Unlike conventional MoE models that apply expert routing only to FFN layers, MICRO routes tokens through entire Transformer blocks, leading to emergent domain-specific experts (e.g., language, logic, social reasoning). The work is conceptually inspired by cognitive neuroscience and provides interesting empirical evidence for modularity in large language models.

**Strengths:**

- The paper is thought-provoking and connects cognitive science with model architecture in an elegant way. The results showing emergent specialization patterns and controllable routing behavior are intriguing. Overall, the idea of modeling “cognitive modularity” within Transformers is both fresh and potentially impactful.

- The paper is generally well written, and the experiments are clearly structured. The figures and analyses are helpful for understanding how modularity emerges under the proposed mechanism.

**Weaknesses:**

- While the modular design is inspired by human cognition, the model enforces top-1 routing per layer, meaning each token is processed by only one expert. In contrast, human reasoning typically involves parallel activation and cooperation among multiple brain regions. This exclusive routing assumption may limit the biological and functional plausibility of the approach.

**Questions:**

No.

---

> ### Author Response · Authors · 2025-11-21
>
> We thank the reviewer for their thoughtful and encouraging feedback, and for recognizing both the conceptual novelty and empirical strength of MiCRo. We especially appreciate the reviewer’s positive remarks about the clarity of the paper and the connection it draws between cognitive neuroscience and model architecture.
>
> Regarding the concern that our modular design enforces exclusive routing (i.e., that each token is processed by only one expert per layer), we would like to clarify that:
>
> 1. **Each token is routed to a potentially different expert at every layer.** While MiCRo uses *top-1 routing* within each layer, the routing decisions are dynamic and token-specific, meaning that a single token is typically processed by multiple different experts across layers within the same forward pass. This means that specialization is layered and cumulative, not static or confined to a single expert pathway.
> 2. **Cross-expert interactions naturally occur through attention.** Tokens processed by different experts still attend to one another through the multi-head attention mechanism. Thus, the model continuously exchanges information across expert boundaries, enabling global integration while preserving local specialization.
> 3. **Top-1 routing was chosen for fair parameter comparison.** We deliberately use top-1 rather than top-k routing to ensure that the number of active parameters per forward pass remains comparable to that of the dense baseline.
>
> We believe this balance between specialization and communication captures the essence of brain-inspired modularity.
>
> That said, following the reviewer’s suggestion, we evaluated one of our models with different values of `k` at inference time and found that our model generalizes well to `k=2` even though the large-scale SFT training phase was done with top-1 routing. Therefore, this can be seen as a form of test-time scaling, by including more experts at each layer at test-time. We include the results for MiCRo-OLMo below and in Appendix L and Table 12 in the revised manuscript.
>
> | Top-K | GSM8K | BBH  | MMLU | MATH  | HellaSwag | PIQA  | ARC Easy | ARC Challenge | Average |
> |-------|-------|--------------|----------------|----------------|------------|-------|----------|----------------|----------|
> | 1     | 45.5% | 29.5%        | 37.9%          | 5.6%           | 65.4%     | 75.2% | 70.2%    | 41.3%          | 46.3%   |
> | 2     | **47.7%** | **30.6%**        | **38.2%**          | **6.8%**           | **66.4%**     | **75.4%** | **71.9%**    | **41.6%**          | **47.3%**   |
>
> We thank the reviewer again for highlighting this important point, which we clarified explicitly in the revised manuscript and included the expert test-time scaling results in Appendix L.

---

> > ### Comment · Reviewer_vTG9 · 2025-11-28
> >
> > Thank you for the detailed clarification and the additional experiments. Your explanation about dynamic routing across layers and cross-expert interaction through attention addresses my original concern well, and the top-k test-time scaling results further strengthen the case.
> >
> > If the review system were still open for score adjustments, I would be happy to raise my rating to 8.

---

### Official Review · Reviewer_1ukn · 2025-11-01

**Soundness:** 3
**Presentation:** 2
**Contribution:** 2
**Rating:** 4
**Confidence:** 3

**Summary:**

This paper introduces MICRO (Mixture of Cognitive Reasoners), a modular transformer architecture inspired by the functional specialization of human brain networks for language, logic, and social reasoning. The model partitions layers of a pretrained language model into four expert modules, each aligned with distinct cognitive domains, and is post-trained using a curriculum to promote specialization. MICRO offers three main advantages: (1) its expert modules are interpretable and causally meaningful, (2) it allows dynamic control at inference by routing tokens to domain-specific experts, and (3) it matches or exceeds baseline performance on reasoning benchmarks (e.g., GSM8K, BBH) and human-alignment tasks (CogBench). Overall, MICRO demonstrates that cognitively inspired modularity can enhance both the interpretability and human-likeness of large language models.

**Strengths:**

The novelty of MICRO lies in its cognitively inspired modular design, where a pretrained language model is partitioned into expert modules aligned with human brain networks for language, logic, and social reasoning. This structure enables interpretable, causally meaningful reasoning and dynamic control at inference, while maintaining or improving performance on both reasoning and human-alignment benchmarks.

The research provides:

•	Interpretable modules that provide causally meaningful insights into model behavior.

•	Dynamic control at inference, allowing selective routing to domain-specific experts.

•	Strong empirical performance, matching or exceeding baselines on reasoning and human-alignment benchmarks.

**Weaknesses:**

There are some shortcomings:

•	Increased model complexity due to multiple expert modules and modular routing.

•	Potential scalability issues for very large models or tasks requiring many cognitive domains.

•	The work makes a valuable contribution and builds effectively on current advances. However, including a discussion of remaining challenges and possible avenues for future research would strengthen the paper and highlight its long-term potential.

**Questions:**

Please discuss limitations of MICRO and how these limitations may be addressed.

It is unclear what is going on in Figs 4 and 16. Please explain the figures.

What is the key for Figure 5?

Please discuss the possible impact of this research.

Please discuss the generalizability of this research.

---

> ### Author Response · Authors · 2025-11-21
>
> We thank the reviewer for their thoughtful and constructive feedback, and for highlighting the novelty of MiCRo in its cognitively inspired modular design and contributions to interpretability, causal insight, and human-alignment. We address the reviewer’s comments below.
>
>
> **Re Weaknesses**
>
> **1/**
> > Increased model complexity due to multiple expert modules and modular routing.
>
> We note that, since MiCRo uses top-1 routing, the number of active parameters at inference is comparable to that of its corresponding baselines. For example, in the Llama-3.2-1B setting, MiCRo and MoB both require 2.8074 GFLOPs per token, while the dense baseline requires 2.4716 GFLOPs per token (the difference being due to the router computation). It is true that both MiCRo and the MoB (Mixture of Blocks) baseline have approximately 4× the total parameter count of the dense model. However, this is precisely why we included MoB as a baseline—to control for the effect of modularity and parameter count. The only difference between MiCRo and MoB lies in the curriculum that induces brain-like specialization. With this factor isolated, we observe that MiCRo not only achieves higher interpretability (as demonstrated through its routing patterns and ablation analyses; Figures 3, 4, 12-14, and 16), but also outperforms MoB for models up to 3Bx4 parameters in downstream benchmarks.
>
> ----------------
>
> **2/**
> > Potential scalability issues for very large models or tasks requiring many cognitive domains.
>
> Following the reviewer’s suggestion, we have post-trained three Llama-3.1-8B variants:
> (1) MiCRo-Llama-8B, (2) its modular baseline Llama-8B-MoB, and (3) Llama-8B-Dense.
> Due to compute and time constraints, we instantiated experts only in the last 12 layers of MiCRo-Llama-8B and Llama-8B-MoB, keeping the earlier layers dense. This choice is motivated by our prior findings (Figures 13–14, Appendix) showing that early layers predominantly route to language experts, while non-language specializations emerge in later layers. The results confirm that MiCRo-Llama-8B exhibits the expected routing patterns in its last 12 layers (Figure 20 in the revised PDF), and we include the results on the reasoning benchmarks in the table below (including MMLU-Pro as per Reviewer A5jV suggestion) and in the revised manuscript (Appendix M).
>
> The table below shows the results on the 9 benchmarks of the three Llama-3.1-8B model variants, along with the results when we remove the most detrimental expert across all layers for a given task for the MiCRo and MoB model variants, which we denote by appending (Ablation), similar to Figure 7 in the main paper. Using a paired Wilcoxon signed-rank test across the nine benchmarks, we find no statistically significant difference between MiCRo and either of the two baselines. The Dense vs. MiCRo comparison yields a test statistic of 31.0 (p = 0.18), and the MoB vs. MiCRo comparison yields 33.5 (p = 0.10), both well above the conventional 0.05 significance threshold. However, when we ablate the most detrimental expert per task for both MiCRo and MoB we see a jump in performance, as also illustrated in Figure 7.
> In general, the Llama-8B experiments confirm our initial results that MiCRo remains competitive on a suite of benchmarks while remaining interpretable and relevant to cognitive neuroscience as discussed in our response to Reviewer D255 (Weakness #4).
>
> | Model Name                | GSM8K | BBH | MMLU | MMLU Pro | MATH  | HellaSwag | PIQA | ARC Easy | ARC Challenge | Average |
> |---------------------------|-------|--------------|----------------|--------------------|----------------|------------|------|----------|----------------|---------|
> | Llama-8B-Dense            | 71.3  | 54.2         | 51.6           | 24.6               | 21.5           | 72.8       | 78.3 | 74.3     | 44.1           | 54.7    |
> | Llama-8B-MoB              | 69.3  | 54.1         | 52.0           | 23.5               | 21.3           | 72.2       | 78.9 | 74.6     | 46.9           | 54.8    |
> | Llama-8B-MoB (Ablation)   | 70.0  | 55.9         | 54.1           | 36.3               | 21.6           | 72.4       | 79.3 | 75.0     | 46.9           | 56.8    |
> | MiCRo-Llama-8B            | 69.1  | 53.6         | 50.7           | 23.1               | 18.1           | 72.7       | 78.7 | 75.7     | 45.9           | 54.2    |
> | MiCRo-Llama-8B (Ablation) | 69.1  | 55.0         | 52.7           | 36.3               | 18.2           | 73.1       | 78.7 | 76.3     | 46.7           | 56.2    |
>
>
> Our open-source code and data includes detailed documentation on how to extend MiCRo to higher parameter counts and additional cognitive domains, enabling the community to explore scalability further. The anonymized repository can be found here: https://anonymous.4open.science/r/mixture-of-cognitive-reasoners-iclr and is included in the revised PDF as a footnote on the first page.
>
> --------------------
>
> The response continues in the next comment.

---

> > ### Author Response · Authors · 2025-11-21
> >
> > **3/**
> > > including a discussion of remaining challenges and possible avenues for future research would strengthen the paper and highlight its long-term potential.
> >
> > We appreciate this suggestion and used the additional page to expand the Discussion & Future Work section (Section 6) as follows in the revised PDF:
> >
> > --------------------------------
> >
> > **Extending Specialization Beyond Cognitive Domains**   While inspired by the brain’s functional organization, our specialization framework can be applied to any meaningful partition of expertise, such as technical domains or natural languages. One key question is whether the model preserves the intended specialization through the large-scale end-to-end training. Our results suggest that brain-inspired partitions provide a robust inductive bias: they persist throughout training and lead to structured, interpretable routing patterns. Supporting evidence in Appendix I shows that expert usage remains consistent across checkpoints over the course of Stage 3 training. Looking ahead, this framework could also be extended to other cognitive domains. For example, recent neuroscience findings point to a distinct brain network involved in abstract formal reasoning such as induction and deduction and another one involved in intuitive physics (Kean et al., 2025a; 2025b). Incorporating a corresponding module could improve the model’s performance on tasks involving such capacities.
> >
> > **Towards Brain Alignment Beyond Language**   Since our model is explicitly designed to mirror distinct cognitive networks in the human brain, and given that established neuroscience localizers can identify the corresponding expert modules, an exciting direction for future work is to examine whether the internal representations of these experts align more closely with neural activity in their respective brain networks (Schrimpf et al., 2018; 2020). Prior studies have shown that language-selective units in large language models correlate more strongly with activity in the human language network than randomly selected units (AlKhamissi et al., 2025a), suggesting a meaningful link between specialization in models and brains. This raises the natural question of whether similar alignment can be observed for other cognitive domains, such as reasoning or social cognition. However, assessing MiCRo’s neural alignment beyond the language network is currently limited by the lack of suitable datasets. Existing fMRI benchmarks rarely engage non-language regions such as the Multiple Demand (Duncan 2010) network and often use blocked designs that preclude item-level analyses---highlighting the need for experimentalists to collect new datasets that explicitly target non-language brain regions. We believe that once suitable neural datasets exist, our model can be used to instantiate specific hypotheses about how these networks—and their corresponding experts—interact and exchange information.
> >
> > **Limitations**   While our approach improves interpretability without sacrificing performance, several open questions remain. Scaling beyond an 8B base model has yet to be demonstrated, and the impact of adding more experts to the current MiCRo architecture is still unknown. The MiCRo_SFT dataset used in Stage-1 (~ 3,000 GPT-4o pseudo-labeled samples) has not been evaluated for size sensitivity, leaving open whether increasing or reducing its scale would alter the degree of specialization or downstream performance. Although \modelname{GPT-4o} provides high-quality pseudo-labels, as demonstrated in Appendix B, using human-annotated data could strengthen the inductive bias and potentially improve the final model. Our post-training pipeline currently includes only SFT and DPO (Appendix G); exploring additional stages such as RLVR remains an avenue for future work. Finally, our evaluation of alignment to human behavior focuses on CogBench, and extending this analysis to a broader set of behavioral or cognitive datasets is an important direction for future research.
> >
> > ---------------------------------
> >
> > Beyond these, we also plan to highlight broader implications, such as the benefits of building brain-aligned models, and the benefits of interpretability for AI safety research.
> >
> > -------------------------------------
> >
> > The response continues in the next comment.

---

> > > ### Author Response · Authors · 2025-11-21
> > >
> > > **Re Questions**
> > >
> > > **1/**
> > > > Please discuss limitations of MICRO and how these limitations may be addressed.
> > >
> > > We acknowledge several current limitations, in addition to the scalability considerations noted above, we have incorporated into the Limitations paragraph (Section 6) in the revised PDF the following points as noted above:
> > >
> > > - The MiCRo-SFT dataset used in Stage-1 (≈3 000 GPT-4o pseudo-labeled samples) has not been tested for size sensitivity due to the large cost of re-running every experiment at different dataset sizes. It remains an open question whether increasing or reducing its scale affects specialization and performance.
> > > - While GPT-4o provided high-quality pseudo-labels for seeding specialization, human-annotated data could further strengthen the inductive bias and improve downstream performance. We expect, however, that the potential of this approach will continue to grow as synthetic labellers become more accurate and reliable, enabling even stronger and more scalable specialization in future versions of this class of methods.
> > > - Our post-training pipeline currently includes SFT and DPO only. Investigating additional steps such as RLVR remains for future work.
> > > - Finally, alignment to human behavior was evaluated on CogBench; future work could extend this analysis to a broader range of behavioral datasets.
> > >
> > > ----------------------------
> > >
> > > **2/**
> > > > It is unclear what is going on in Figs 4 and 16. Please explain the figures.
> > >
> > > Figures 4 and 16 present performance deltas between the intact model and versions with individual experts ablated across all layers. Each bar represents the performance drop (or gain) after removing a specific expert, thus quantifying causal specialization.
> > > For example, in GSM8K (math word problems), removing the logic expert causes a ~ -28 % drop (from 34.72 %), whereas ablating the social expert slightly improves performance (~ +5 %), confirming that each expert contributes selectively to different domains. Figure 16 similarly shows that the language expert is essential across all benchmarks.
> > >
> > > ----------------------------
> > >
> > > **3/**
> > > > What is the key for Figure 5?
> > >
> > > The key is a color scale from dark blue to light yellow representing the percentage of neurons within each expert (per layer) that belong to the top 10 % of active parameters of the entire model, as defined by the corresponding functional localizer. This mirrors how neuroscientists identify brain networks (e.g., the human language network; Lipkin et al., 2022). We added a color bar legend to each plot in the revised version for clarity.
> > >
> > > ----------------------------
> > >
> > > **4/**
> > > > Please discuss the possible impact of this research.
> > >
> > > As expanded in Discussion & Future Work, the long-term goal of MiCRo is to advance the development of brain-aligned models—systems whose modular organization reflects human cognition while maintaining high task performance and behavioral alignment. This approach not only enhances interpretability and control but also enables digital-twin applications, allowing cognitive neuroscientists to test hypotheses in silico before experimental validation. MiCRo represents a concrete step toward such neuro-computational integration.
> > >
> > > ----------------------------
> > >
> > > **5/**
> > > > Please discuss the generalizability of this research.
> > >
> > > We validated our post-training framework across six base models of different sizes and families, demonstrating that MiCRo’s specialization procedure generalizes robustly across architectures. As discussed above, further work will explore generalization under larger scales and additional cognitive domains.
> > >
> > > ----------------------------
> > >
> > > In summary, we appreciate the reviewer’s constructive suggestions, which we believe further strengthened the revised manuscript by clarifying figures, explicitly stating limitations, and expanding the discussion on impact and future directions.

---

> > > > ### Comment · Reviewer_1ukn · 2025-11-25
> > > >
> > > > I sincerely appreciate the clarification that the authors have highlighted in their responses. They strengthen the paper. My main concern is that there is still a lot of work that could be done to strengthen the paper, as the authors have highlighted in their comments. These extensions could be the focus of further papers. I am happy to raise my grade given the changes to the current paper that the authors have indicated.

---

### Official Review · Reviewer_JAhk · 2025-11-02

**Soundness:** 4
**Presentation:** 4
**Contribution:** 3
**Rating:** 8
**Confidence:** 3

**Summary:**

The paper proposes the Mixture of Cognitive Reasoners, a modular, transformer-based architecture that built based on neuroscience background with individual language, logic, social, and world experts. The training procedure of the model is composed by the following three stages: Training the experts with a small curated dataset to provide basic inductive bias; Training the router using the same dataset, with the experts frozen; Training the whole model with large instruction-tuning dataset. Experiments shows that the model acquires experts specialization, exhibits alignment to human behaviors, and matches dense and non–brain-aligned modular baselines on several tasks.

**Strengths:**

- The paper is well-written and understandable.
- The neuroscience motivation of the architecture design is very novel.
- The training procedure is novel.
- The experimental results are promising, showing that the proposed Mixture of Cognitive Reasoners model lead to strong interpretability and performance gain over other architectures.

**Weaknesses:**

- The language, logic, social, and world experts decomposition might not be optimal.

**Questions:**

I do not have additional questions at this stage.

---

> ### Author Response · Authors · 2025-11-21
>
> We thank the reviewer for their encouraging feedback, and for recognizing both the neuroscience motivation of MiCRo’s architecture and the novelty of its training procedure for inducing brain-like specialization. We also appreciate the reviewer’s acknowledgment of our promising empirical results demonstrating that cognitively inspired modularity can yield both interpretability and competitive performance.
>
> Regarding the reviewer’s comment on the potential optimality of the expert decomposition, we agree that other partitions may exist beyond the current language, logic, social, and world experts. For example, the MoB baselines we compare our MiCRo models against—both of which have the exact same architecture—do learn another subdivision on their own as we do not force any initial inductive bias. Despite that, MiCRo does outperform the MoB for models up to 3Bx4 with this cognitively inspired expert decomposition (39.4% vs 37.4% on average for the Llama-3.2-3B base mode for example; Figure 7). That said, our choice of this particular decomposition was guided by three main considerations:
>
> 1. **Domain coverage**: these four domains span a broad diverse range of cognitive and reasoning functions typically involved in language model tasks.
> 2. **Architectural simplicity**: limiting the number of experts prevents redundancy and ensures that each expert develops a distinct and meaningful specialization.
> 3. **Neuroscientific grounding**: the selected domains correspond to well-established functional networks in the human brain, namely, the Language Network, Multiple-Demand Network, Theory-of-Mind Network, and Default Mode Network.
>
> We see this as a starting point rather than a limitation. Future work could explore alternative decompositions, for example, introducing additional modules inspired by neuroscience evidence (e.g., for intuitive physics or affective reasoning).

---

### Author Response · Authors · 2025-12-02
**General Comment**

## General Comment

We thank all reviewers for the thoughtful evaluations. Below we briefly summarize the main contributions of the paper, the main concerns raised prior to rebuttal, how they were addressed, and any follow-up clarifications made during discussion.

---

## Paper Summary

This paper presents Mixture of Cognitive Reasoners (MiCRo), a modular transformer architecture inspired by functional specialization in the human brain. A pretrained language model is split into four expert modules (Language, Logic/MD, Social/ToM, World/DMN) and post-trained with a three-stage curriculum to induce stable specialization. The result is a model whose experts are interpretable, causally functional, and controllable at inference via expert routing. MiCRo maintains competitive reasoning performance versus dense and modular baselines, and in ablation settings can exceed them, while also showing improved alignment to human behavioral patterns (CogBench). The core contribution is demonstrating that cognitively grounded modularity can produce human-interpretable experts without hurting capability, offering a pathway toward more transparent and controllable LLMs.

---

## Rebuttal Outcome

During rebuttal, we addressed all concerns with new experiments (scaled up to 8B + evaluation on MMLU-Pro making 9 ML benchmarks in total + self-consistency compute control + test-time scaling results by allocating more experts at test-time), added human validation for the pseudo-labels with stats, replicated all figures for the new 8B model (e.g., routing patterns), clarified figures and neuroscience framing, and expanded the discussion of limitations. Reviewers expressed increased confidence, three raised or intended to raise their scores, and no unresolved concerns remain.

---

## Reviewer JAhk
**Main Concern**: Whether the chosen four-expert decomposition (language/logic/social/world) is optimal.

**Our Clarification**: We acknowledged alternative decompositions may exist but showed that (i) MiCRo outperforms the non-specialized baselines (i.e., MoB) at equal routing/compute under this partition, (ii) choices were neuroscience-motivated, minimally redundant, and covering a diverse set of cognitive and reasoning functions, and (iii) future work can extend to additional experts (e.g., intuitive physics).

**Outcome**: Reviewer expressed full satisfaction and had no further questions.

---

## Reviewer 1ukn
**Main Concerns**: Requested clearer limitations, clarification of Figs. 4/5/16, scaling issues, and a discussion on impact & generalizability.

**Our Clarification**:
- Extended Limitations + Future Work section in revision to talk about generalizability and long-term impact of this work
- Added explicit figure legends & explanation of ablation deltas
- Trained an 8B model along with its two corresponding baselines
- Included evaluation at 8B scale + MMLU-Pro for all other models, as per reviewer suggestion

**Outcome**: Reviewer upgraded from 4 → 6, explicitly acknowledging improved clarity.

---

## Reviewer vTG9
**Main Concerns**: Biological plausibility of specialized experts given top-1 routing inference.

**Our Clarification**: : Routing is dynamic per-token and varies across layers; cross-expert attention allows information exchange. We also ran top-k inference experiments showing MiCRo generalizes well to k>1 even without re-training.

**Outcome**: Reviewer stated they would raise score to 8 if updates were still open.

---

## Reviewer A5jV
**Main Concerns**: Scale beyond 3B, MMLU-Pro inclusion, compute comparison for ablation.

**Our Clarification**:
- Added new experiments at 8B, verifying specialization and competitiveness
- Added MMLU-Pro results to main + appendix
- Ran self-consistency 3-vote baseline to control for test-time compute → MiCRo-Ablation remains stronger

**Outcome**: Reviewer noted satisfaction with results and discussion resolution but didn’t have time to respond to our final set of experiments.

---

## Reviewer D255
**Main Concerns**: Neuroscience mapping simplification, validity of the MiCRo_SFT dataset pseudo-labels used to induce specialization, interpretability claims, cognitive relevance.

**Our Clarification**:
- Emphasized coarse neuroscientific grounding, and that our aim was to test whether ideas proposed in the cognitive neuroscience literature can provide a useful inductive bias for modular learning in LLMs
- Added human validation for the MiCRo_SFT dataset (n=253 w/ agreement stats)
- Reframed interpretability as causal controllability and domain-specialized routing
- Highlighted concrete neuroscientific hypotheses & experimental predictions

**Outcome**: Reviewer explicitly stated it was one of the best papers they reviewed, agreed with clarifications, and mentioned they will update the score.

---

> ### Author Response · Authors · 2025-12-02
> **General Comment Cont.**
>
> ## Overall
>
> All substantive issues: scalability, human labeling, figure clarity, neuroscience framing, and compute fairness, were directly addressed with new experiments, new analysis, and manuscript updates. Reviewer A5jV was not able to provide a final follow-up after we completed the requested additional experiments they raised. However, three other reviewers explicitly reported increased confidence and that they either increased or were planning on increasing their final score.

---

### Meta-Review · Area_Chair_9k1a · 2026-01-07

**Summary:**

The paper suggests MiCRo - Mixture of Cognitive Reasoners, a transformer architecture with four designated "experts" inspired by cognitive science and trained for specialization via supervision. It is claimed to enhance interpretability, control and performance.

Reviewers found the cognitive-inspired design interesting and thought provoking, and were also positive about the clear presentation. Concerns were raised about the method's complexity, scalability potential, and several aspects of the experimental design and comparisons to baselines. The authors responded comprehensively, and it seems the sentiment among the reviewers was ultimately more positive than the raw scores let on.

Even after the authors' responses and additions, there are quite a few lingering methodological questions and concerns about the paper. However, none seems disqualifying, and there will always be some in research of this kind. Given the inspired nature of the paper and the interest and discussion it generated, I believe the ICLR program may benefit from it, and recommend acceptance.

**Note on CoE:** The paper describes experiments with human-annotated data but is vague about whether the data was collected as part of the described research. The rebuttal explicitly describes a new experiment that involved human annotators. The ICLR CoE states that research involving human annotators should assess and report the need for ethical approvals. After bringing this up with the PC, the authors are asked to refer to the CoE and include the missing disclosure in the revised paper.

**Reviewer Concerns:**

Method's complexity, scalability potential, and several aspects of the experimental design and comparisons to baselines.

**Reviewer Scores:**

Most reviewers got to respond before discussions were halted and in most cases their views on the paper seem to have become more positive.

---

### Decision · Program_Chairs · 2026-01-26

Accept (Poster)